# Concerted modification of nucleotides at functional centers of the ribosome revealed by single-molecule RNA modification profiling

Andrew D Bailey[1]*[†], Jason Talkish[2][†], Hongxu Ding[1,3], Haller Igel[2], Alejandra Duran[4], Shreya Mantripragada[5], Benedict Paten[1]*, Manuel Ares[2]*

[1]Department of Biomolecular Engineering and Santa Cruz Genomics Institute, University of California, Santa Cruz, Santa Cruz, United States; [2]RNA Center and Department of Molecular, Cell & Developmental Biology, University of California, Santa Cruz, Santa Cruz, United States; [3]Department of Pharmacy Practice & Science, College of Pharmacy, University of Arizona, Tucson, United States; [4]Colegio Santa Francisca Romana, Bogotá, Colombia; [5]Monta Vista High School, Cupertino, United States

*For correspondence:
bailey.andrew4@gmail.com (ADB);
bpaten@ucsc.edu (BP);
ares@ucsc.edu (MA)

[†]These authors contributed equally to this work

Competing interest: The authors declare that no competing interests exist.

## Abstract

Nucleotides in RNA and DNA are chemically modified by numerous enzymes that alter their function. Eukaryotic ribosomal RNA (rRNA) is modified at more than 100 locations, particularly at highly conserved and functionally important nucleotides. During ribosome biogenesis, modifications are added at various stages of assembly. The existence of differently modified classes of ribosomes in normal cells is unknown because no method exists to simultaneously evaluate the modification status at all sites within a single rRNA molecule. Using a combination of yeast genetics and nanopore direct RNA sequencing, we developed a reliable method to track the modification status of single rRNA molecules at 37 sites in 18 S rRNA and 73 sites in 25 S rRNA. We use our method to characterize patterns of modification heterogeneity and identify concerted modification of nucleotides found near functional centers of the ribosome. Distinct, undermodified subpopulations of rRNAs accumulate upon loss of Dbp3 or Prp43 RNA helicases, suggesting overlapping roles in ribosome biogenesis. Modification profiles are surprisingly resistant to change in response to many genetic and acute environmental conditions that affect translation, ribosome biogenesis, and pre-mRNA splicing. The ability to capture single-molecule RNA modification profiles provides new insights into the roles of nucleotide modifications in RNA function.

## Editor's evaluation

The paper describes a method for single-molecule profiling of RNA modifications. The results not only solve many urgent questions in understanding rRNA modification, ribosome heterogeneity and ribosome biogenesis, they also provide a major step in developing technologies to probe the RNA epitranscriptome. The results are expected to be of broad interest for specialists in the RNA field.

## Introduction

In addition to the four standard nucleotides, there are more than 160 distinctly modified ribonucleotides and more than 50 distinctly modified deoxyribonucleotides found in the RNA and DNA of cells (*Boccaletto et al., 2018*; *Jonkhout et al., 2017*; *Sood et al., 2019*). Many of these modified

nucleotides provide extra regulatory information and are crucial for cell function. Irregular DNA methylation patterns are linked to several cancers, neurological disorders and autoimmune diseases (*Portela and Esteller, 2010*; *Raiber et al., 2017*). Aberrant RNA modification has been linked to the development of cognitive functions, neurological defects, breast cancer, genetic birth defects, and diabetes (*Bednářová et al., 2017*; *Benrahma et al., 2014*; *Delaunay et al., 2016*; *Delaunay and Frye, 2019*; *Hideyama et al., 2012*; *Wu et al., 2005*). In ribosomal RNA (rRNA), extensive and highly conserved modifications are vital for correct ribosome structure and function (*Polikanov et al., 2015*; *Sloan et al., 2017*). Modifications on rRNA have been generally considered to be constitutive in support of fine tuning function (*Karijolich et al., 2010*) rather than mediating specific regulatory changes in ribosome function. However, the fraction of rRNA molecules modified at individual positions can change in response to the environment, disease and developmental state (*Birkedal et al., 2015*; *Sloan et al., 2017*; *Taoka et al., 2016*), and evidence is accumulating that ribosome heterogeneity may influence translation (*Gay et al., 2022*; *Liu et al., 2021b*). It seems possible that modification status throughout locations in the ribosome could control translation by creating functional heterogeneity in the cell's pool of ribosomes.

One technical challenge of analyzing the effect of modification on the function of RNA is that no method is available to capture the modification state at all positions of a single RNA molecule. Traditional approaches examine ensembles of molecules and estimate the fraction modified at individual sites independently. For example, non-sequencing based techniques such as liquid chromatography-tandem mass spectrometry (LC-MS/MS) (*Taoka et al., 2009*) and cryogenic electron microscopy (cryo-EM) (*Natchiar et al., 2017*) can identify the presence of all types of modified nucleotides in ensembles of rRNA molecules. Some methods such as immunoprecipitation-seq (*Dominissini et al., 2012*) or mismatch-seq (*Ramaswami et al., 2013*), aggregate information from several reads to detect modifications at a specific site, but do not capture associations between modification status at distant sites in large RNA molecules. Other approaches such as bisulfite-seq (*Frommer et al., 1992*), Ψ-seq ('psi-seq') (*Schwartz et al., 2014*), and RiboMeth-seq (*Birkedal et al., 2015*) are highly specific for a single type of modified nucleotide, but also require fragmentation of RNA, preventing capture of modification status at multiple distant sites in an RNA. Such whole molecule information is necessary to assess the relationship between function and modification status of individual ribosomal subunits.

New advances in direct single-molecule sequencing of RNA using nanopore technology may circumvent many of these limitations. Direct nanopore sequencing of full-length RNA molecules (*Deamer et al., 2016*; *Garalde et al., 2016*) has the potential to report modification status across entire RNA molecules without chemical treatment or amplification steps. Modified nucleotides produce changes in electrical current distinct from canonical nucleotides, permitting modification detection algorithms to identify modifications in both DNA and RNA. Given enough training data, basecalling algorithms can predict both canonical nucleotides and modifications directly from the signal (*Wick et al., 2019*). However, training data for most modifications is limited, and thus many algorithms rely on aligning reads to a reference sequence to identify modified nucleotides (*Furlan et al., 2021*). Current signals can be modeled using secondary features like quality scores and base miscalls (*Begik et al., 2021*; *Liu et al., 2021a*) or directly using the underlying signal (*Rand et al., 2017*; *Simpson et al., 2017*; *Stoiber et al., 2016*). However, no currently available method captures combinations of distinctly modified nucleotides at multiple distant sites in RNA.

Here, we demonstrate accurate, single molecule modification profiling of 13 distinct types of modified nucleotides at 110 positions across full transcripts of 18 S and 25 S rRNA from *S. cerevisiae*. We preserve long-range associations between modification status at distant positions on single RNA molecules, allowing us to identify highly correlated positions and explore heterogeneity in ribosomal RNA modification. Clustering analysis separates populations of distinctly modified ribosomes in wild type yeast, as well as in yeast deficient in gene functions required for modification including Cbf5, Nop58, several snoRNAs, RNA helicases Dbp3, Prp43, and their G-patch protein partners. These studies provide evidence that groups of nucleotides are modified in a concerted manner, especially at functional centers of the ribosome. Further application of single-molecule modification profiling will enable dissection of the contributions of nucleotide modification to the function of large RNAs.

# Results

## Profiling rRNA Modifications at Single Molecule Resolution

To investigate the overall modification status of yeast rRNA on a single molecule level, we used nanopore current traces from Oxford Nanopore MinION flow cells (see Materials and methods) of complete rRNA transcripts to capture modification status at every modified position along individual molecules. To create these single molecule profiles, we trained signalAlign (*Rand et al., 2017*) by modeling wild type rRNA reads as 'modified' and in vitro transcribed (IVT) reads as 'unmodified' to detect all 110 annotated modifications in *S. cerevisiae* 18 S and 25 S rRNA (*Figure 1—figure supplement 1* and file supplement 1) (see Materials and methods) (*Stoiber et al., 2016*; *Taoka et al., 2016*). For each rRNA read, the model estimates the probability of modification, regardless of modification type, at each annotated position and outputs a list of modification probabilities for each full-length rRNA read (*Figure 1—figure supplement 2A*).

We wondered whether different genetic and environmental conditions would alter modification profiles to reveal distinct subpopulations of ribosomes. To test the ability of the trained model to capture single-molecule modification profiles, we examined yeast suffering catastrophic loss of snoRNA-guided rRNA modifications, by depleting either the C/D box (2′O-methylation) or H/ACA box (pseudouridylation) class of snoRNPs. In *S. cerevisiae*, 34 of the 37 18 S and 66 of the 73 25 S rRNA annotated modifications are guided by the C/D box (2′O-methylation) and H/ACA box (pseudouridylation) snoRNPs (*Piekna-Przybylska et al., 2007*; *Tollervey and Kiss, 1997*; *Yang et al., 2016*). To ablate these modifications *en masse* we used strains in which Nop58 (core component of C/D snoRNPs) or Cbf5 (H/ACA snoRNP pseudouridylase) can be depleted using a *GAL1* promoter (*Lafontaine and Tollervey, 1999*; *Watkins et al., 1998*). Thus, in galactose-grown cells shifted to glucose medium, Nop58 (or Cbf5) expression will be repressed, leading to substantial loss of functional C/D box (or H/ACA box) snoRNPs, thus blocking modification (*Lafontaine et al., 1998*; *Lafontaine and Tollervey, 1999*). Under these conditions, single-molecule modification profiles produced by our model reveal accumulation of large numbers of rRNA molecules lacking most 2′O-methyl (Nm) (Nop58-depleted) or pseudouridine (Ψ) (Cbf5-depleted) modifications at snoRNA-guided positions in 18 S (*Supplementary file 1B Figure 1—figure supplement 3*) or 25 S rRNA (*Figure 1*).

To examine subpopulations of modified rRNA molecules in these cells, we performed hierarchical clustering. After pooling profiles from the separate samples and clustering them, we observe clear separation of subpopulations representing wild type rRNA, IVTs, and molecules arising from the Nop58 or Cbf5-depleted cells (*Figure 1A* and *Figure 1—figure supplement 3A*). We used dimension reduction UMAP visualization (*McInnes et al., 2018*) of 18 S and 25 S rRNA modification profiles to confirm the presence of these distinct molecular populations (*Figure 1C* and *Figure 1—figure supplement 3C*). Comparing clusters 1 and 3 derived from cells depleted for either of the two different classes snoRNPs (*Figure 1A and C*) shows very little overlap in rRNA modification profiles, suggesting that 2′O methylation and pseudouridylation are largely independent of each other. Some molecules from snoRNP-depleted cells appear to be modified normally and are found in cluster 2 with wild type rRNA (*Figure 1A and C*), more often for 18 S rRNA than for 25 S rRNA (compare to *Figure 1—figure supplement 3*). This may reflect a more severe impact of modification loss on accumulation of 18 S rRNA (*Lafontaine et al., 1998*; *Lafontaine and Tollervey, 1999*), leading to higher residual ribosomes modified before depletion was complete. We conclude that clustering of single-molecule rRNA modification profiles reveals two large but distinct classes of undermodified rRNA molecules induced by depletion of each of the two main classes of snoRNPs.

A powerful advantage of single-molecule modification profiling is the ability to measure concerted changes between modifications within the same RNA. To test this, we measured the change in Spearman correlation between pairs of modified sites, in the absence of Cbf5 or Nop58, compared to wild type rRNA (Fisher z-transform test and Brown's method, see Materials and methods) (*Figure 1D and E*, *Figure 1—figure supplement 3E, F* and *Supplementary file 1A*). In the absence of Cbf5 and Nop58, we observe that the correlated loss of pseudouridine (p-value = 5.5e-05, Brown's method) and 2′O-methyl (p-value = 1.5e-16, Brown's method) modifications is highly significant, respectively (*Figure 1D and E* and *Figure 1—figure supplement 3E,F*). As suggested by the clustering results, a comparison of the pairwise correlation tests for all combinations of modified positions confirmed that to a large extent 2′O-methylation and pseudouridylation in yeast are independent of each other (*Figure 1D and E*).

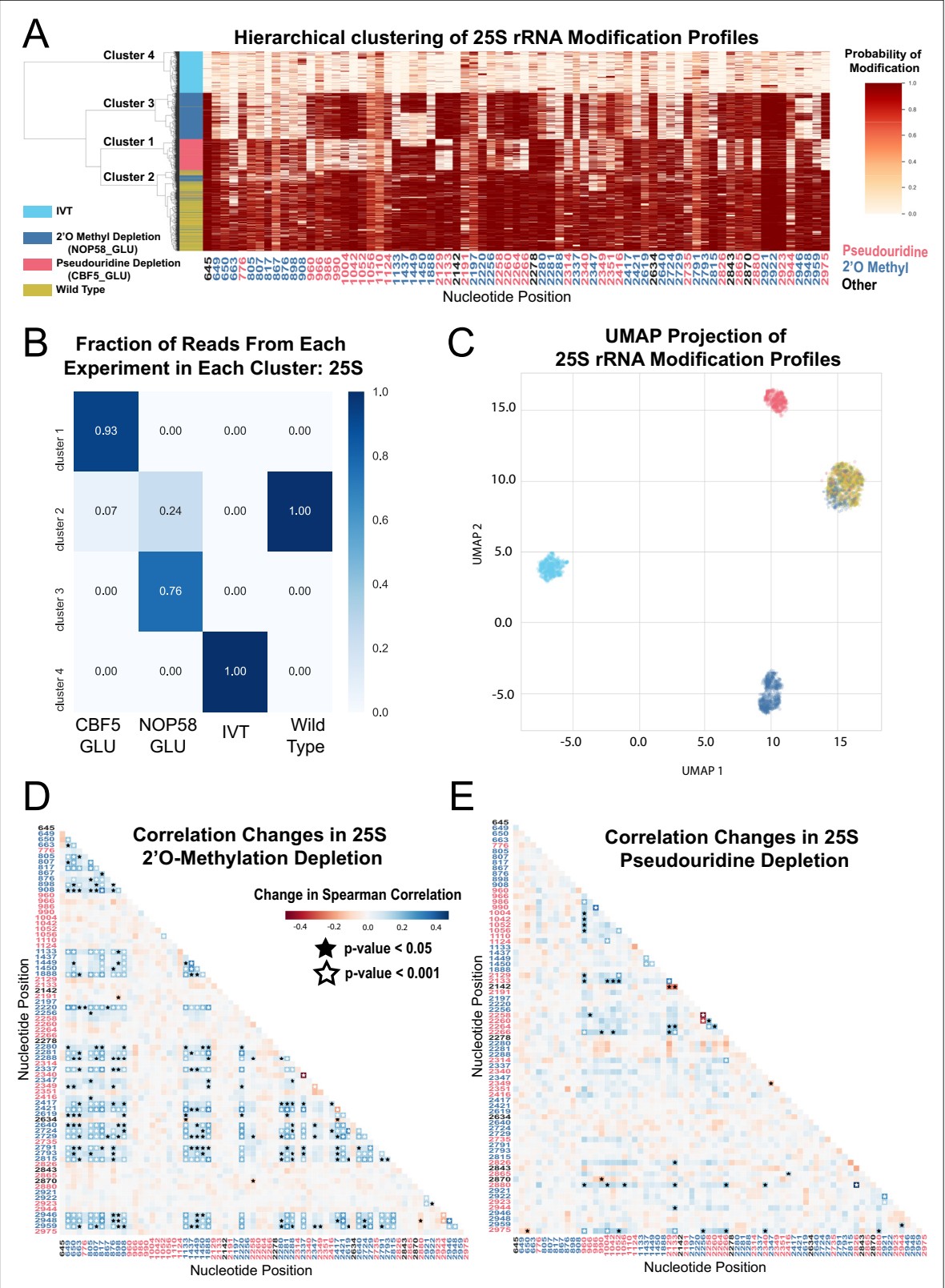

**Figure 1.** Clustering and correlation analysis of depletion experiment modification profiles in 25 S. (**A**) Hierarchical clustering of 25 S yeast rRNA modification profiles of IVT, wild type, and both pseudouridine and 2'O methyl depletion experiments (1,000 reads/experiment). Each row is a full-length molecule, each column is a modified nucleotide and the color represents modification probability, see scale. (**B**) Fraction of reads from IVT, wild type and both depletion experiments in each cluster of 25 S rRNA. (**C**) UMAP visualization of 25 S yeast rRNA modification profiles of IVT, wild type, and both

*Figure 1 continued on next page*

*Figure 1 continued*

pseudouridine and 2′O methyl depletion experiments. UMAP color scheme is the same as the labels in panel A. (**D,E**) Change in Spearman correlations of 25 S reads in 2′O methyl depletion (**D**) and pseudouridine depletion (**E**) when compared to wild type. Stars represent significant changes when compared to wild-type correlation (see Materials and methods). Nucleotide positions are blue for 2′O-methyl, red for pseudouridine, and black for other (neither 2′O-methyl nor pseudouridine).

The online version of this article includes the following figure supplement(s) for figure 1:

**Figure supplement 1.** De-novo detection of modifications using Tombo.

**Figure supplement 2.** SignalAlign pipeline overview, overall accuracy metrics from testing data and per-position model accuracy.

**Figure supplement 3.** Clustering and correlation analysis of depletion experiment modification profiles in 18 S rRNA.

**Figure supplement 4.** Clustering of underlying events to search for patterns of modification in the pseudouridine and 2'O methyl depletion experiments.

**Figure supplement 5.** Analysis of yeast rRNA modification frequency in relation to functional centers of the ribosome.

Several subpopulations of molecules displayed unusual patterns of modification. The sites of methylation guided by the C/D box snoRNA U24 (Cm1437, Am1449, and Gm1450) within the polypeptide exit tunnel (PET) in 25 S rRNA appear to be modified in a concerted fashion. Approximately 40% of the molecules from the Nop58 depletion remain methylated at all three sites despite being unmethylated at most other sites, (*Figure 1A*), suggesting that either the modification of this triad is more efficient, or that rRNAs lacking these modifications are more unstable. Furthermore, about 13% of molecules in the cluster formed by depletion of the pseudouridylase Cbf5 are also unmethylated at all three sites, suggesting that concerted methylation at these positions may be partly dependent on pseudouridine modification. Particularly striking is the highly concerted modification between 25 S rRNA positions Um2921 with Gm2922 and Ψ2923 in the peptidyl transfer center (PTC). These appear refractory to loss of modification in both depletion experiments, remaining modified on a large fraction of molecules otherwise lacking multiple other modifications (*Figure 1A*). Modification of Um2921 is guided by C/D box snoRNA snR52, and Gm2922 is modified by the non-snoRNP methyltransferase Spb1, which can also methylate Um2921 in the absence of snR52 (*Lapeyre and Purushothaman, 2004*). This suggests these modified nucleotides are so crucial that cells have evolved redundant pathways to ensure their modification. Consistent with this, methylation of G2922 appears to be critically required for maturation and nuclear export of the pre-60S ribosomal subunit (*Yelland et al., 2022*). However, modification of Ψ2923 does not have a known redundant snoRNP or enzymatic backup mechanism for modification in the absence of snR10. Thus, the extremely low number of observed molecules lacking modification at these important positions suggests that rRNAs do not accumulate efficiently in their absence.

We also observe a concerted change in loss of modification between the two N4-acetylcytidines (ac⁴C1280 and ac⁴C1773) in 18 S, and loss of 2′O-methylation at many positions in the Nop58 depletion (*P*-value = 2.3e-05, Brown's method) (*Figure 1—figure supplement 3A,E*). N4-acetylcytidine modification depends on the C/D box snoRNAs snR4 and snR45, which are not known to guide methylation, but instead bring the cytidine acetylase Kre33 to positions C1280 and C1773, respectively (*Sharma et al., 2017*; *van Nues et al., 2011*). These atypical C/D box snRNAs also require Nop58, explaining the coordinate loss of cytidine acetylation and 2′O-methylation. We confirmed that our model recognizes ac⁴C modified sites by knocking out snR4 and snR45 and observing loss of the expected N4-acetylcytidine (*Figure 2—figure supplement 1A,B*). Also, the N1-methyl-N3-aminocarboxypropyl-pseudouridine (m¹acp³Ψ1191) residue in 18 S is significantly correlated with pseudouridine positions in the Cbf5 depletion (p-value = 5.4e-07, Brown's method) (*Figure 1—figure supplement 3A,F*), as expected given that snR35-guided pseudouridylation of U1191 is the first step to generate this complex modification (*Brand et al., 1978*).

The concerted modifications in the peptidyl transferase center (PTC) (Um2921, Gm2922, Ψ2923) and the polypeptide exit tunnel (PET) (Am1449, Gm1450) occur on sequential nucleotides at each location. Given that our model incorporates information from sequential 5-mers that contain the position under consideration, pairs of modifications close in sequence share 5-mers, presenting a challenge to the prediction algorithm. To examine the extent to which the model may be affected by closely spaced modifications in these regions, we examined the underlying current signal directly, by clustering the single molecule signal means (see Materials and methods) (*Ding et al., 2020*). This test

reveals that the concerted pattern of modifications predicted by the model agree with the underlying event means clustering, with the exception that modification of $\Psi$2923 may be slightly overestimated (*Figure 1—figure supplement 4* and note supplement 1). We conclude that single-molecule modification profiling allows identification of subpopulations of individual rRNA molecules and captures concerted patterns of modification at functionally important sites in the ribosome.

## Resolving subpopulations of ribosomes that differ at a single modified site

The global loss of modification by depletion of snoRNPs creates grossly undermodified rRNA molecules that are easily distinguished by profiling. To test the sensitivity of the method to resolve classes of ribosomes that differ by a single modification, we first estimated variation in wild-type rRNA profiles that could arise from a combination of experimental noise, behavior of the model, or biological variation in modification. For experiments where capture of a few thousand molecules in a test mutant or condition is sufficient for profiling, we used Flongle flow cells to acquire on average ~6000 full-length 18 S and 25 S rRNA reads. We calculated the variance of the predictions for each annotated modification for each of three wild type, biological replicates. Based on the largest variance (position 562 in 18 S, ~ 9%), we chose a conservative cutoff of a 10% change in modification to call any site affected by a given experimental perturbation (mutation or treatment, *Figure 2—figure supplement 1*). We consider this a very conservative threshold given that many sites were never observed to vary more than a few percent in any experiment. We also compared the predicted modification frequency at a given site in an experiment to its predicted frequency in wild type using a chi-square test (see Materials and methods, *Supplementary file 1B*). We sequenced rRNAs from strains containing individual snoRNA knockouts (snR80, snR83, snR87, snR4, and snR45) previously shown to completely lack modification at one or a few annotated sites in each case (*Piekna-Przybylska et al., 2007*), demonstrating significant decreases in modification at the appropriate site for each (p-values < 1e-04, chi-square test) (*Figure 2—figure supplement 1B and E*). As expected, signal distributions from snoRNA knockout kmers match the model's canonical unmodified kmer distributions (*Figure 2—figure supplement 2*). This experiment confirms our ability to identify loss of modification at single locations with high confidence.

To test the limits of our ability to deconvolute a heterogeneous mixture of ribosomes differing by single modifications, we pooled equal amounts of total RNA from three snoRNA knockout strains (snR80, snR83, and snR87) and wild type, and acquired single molecule modification profiles from the mixture (*Figure 2*). Hierarchical clustering of the profiles obtained from the mixture reveals four similarly sized main clusters of differently modified 18 S rRNA (*Figure 2A and C*) (see Materials and methods). We observe positive correlation changes between positions $\Psi$1290 and $\Psi$1415 (*Figure 2B*) (p-value = 6.4e-16, Fisher z-transform test), that likely arise from their shared dependence on snR83. Moreover, we observe negative correlation changes between all independent pairs of modification sites at Am436, $\Psi$759, and [$\Psi$1290+ $\Psi$1415] only in the artificial mixture (*Figure 2B* and *Supplementary file 1A*) (p-value = 3.5e-08, Brown's method). Since loss of modification at these positions in this test mixture arose independently in the separate samples that were mixed, the population would be extremely unlikely to have molecules with simultaneous loss of modification at any pair of these three locations, creating the negative correlation change. Together, this analysis establishes the ability of the method to resolve minority subpopulations of ribosomes with subtly different modification profiles.

Having observed correlated changes in modification within an artificial mixture of snoRNA knockouts, we asked if modifications not directly guided by the deleted snoRNA change in a concerted fashion in the absence of a particular snoRNA. We found significant and reciprocal changes in correlation in the absence of snR83 and snR4. Upon depletion of snR83 and loss of 18S-$\Psi$1290, position ac$^4$C1280 (snR4/Kre33) shows a negative correlation change (*Figure 2—figure supplement 3A*), consistent with an increase in modification of ac$^4$C1280 by ~9.2% when snR83 is absent. This suggests that the snoRNPs that guide $\Psi$1290 (snR83) and ac$^4$C1290 (snR4/Kre33) modifications may compete for binding of the two closely spaced positions in which the binding sites of the two snoRNAs are predicted to overlap (*Figure 2—figure supplement 3C,D*; *Piekna-Przybylska et al., 2007*; *Schattner et al., 2004*; *Sharma et al., 2017*). In a reciprocal fashion, deletion of snR4 and loss of ac$^4$C1280 produces a negative correlation change with position $\Psi$1290 (*Figure 2—figure supplement 3B*). However, the modification status of $\Psi$1290 does not appreciably change relative

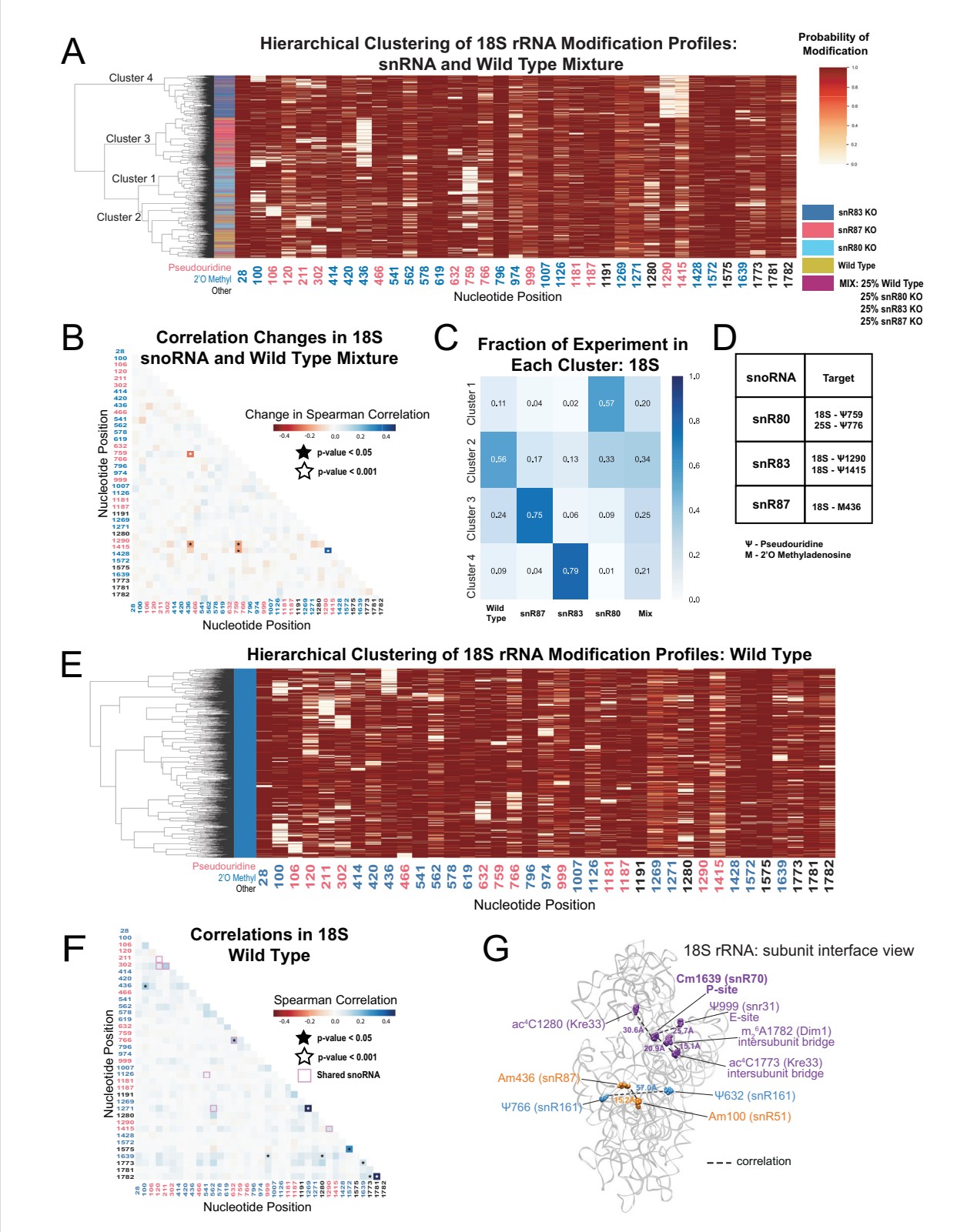

**Figure 2.** Clustering of 18 S rRNA modification profiles and correlation analysis from the mixture experiment and wild type rRNA. (**A**) Hierarchical clustering of 18 S modification of profiles from pooled wild type, snR80Δ, snR83Δ, and snR87Δ RNA (500 reads in each experiment). Each row is a full-length molecule, each column is a modified nucleotide and the color represents modification probability, see scale. (**B**) Change in Spearman correlations of 18 S reads in the mixture experiment when compared to wild type. Stars represent significant changes when compared to wild-type correlation and

*Figure 2 continued on next page*

*Figure 2 continued*

significantly different from zero correlation (see Materials and methods). (**C**) Fraction of wild type, snR80Δ, snR83Δ, and snR87Δ profiles in each cluster of 18 S rRNA. (**D**) Table of snoRNAs knocked down with the corresponding expected knocked down modifications. (**E**) Hierarchical clustering of 18 S yeast rRNA modification profiles from wild-type yeast (1000 reads). (**F**) Wild-type Spearman correlation of 18 S wild-type reads. Stars represent significantly different to IVT correlations and significantly different from zero. correlation. (**G**) Crystal structure model of wild-type *S. cerevisiae* 18 S rRNA highlighting significant correlated positions. PDB: 4V88 (*Ben-Shem et al., 2011*).

The online version of this article includes the following figure supplement(s) for figure 2:

**Figure supplement 1.** Heatmaps and percent modification change of snoRNA knockout and mixture experiments.

**Figure supplement 2.** Kmer distribution comparison between snoRNA knockout kmer distributions and the trained model kmer distributions.

**Figure supplement 3.** Clustering and correlation analysis of snoRNA KO experiment modification profiles in yeast 18 S rRNA.

**Figure supplement 4.** Comparison of rRNA 2'O-methylation calling from other modification detection techniques and signalAlign modification detection.

**Figure supplement 5.** Yeast 25 S rRNA modification profile clustering and correlation analysis.

to wild type (*Figure 2—figure supplement 3D*). These results suggest a hierarchy in which $\Psi 1290$ is modified by snR83 snoRNP prior to modification of ac$^4$C1280, and in the absence of snR83 or $\Psi 1290$, modification of ac$^4$C1280 occurs more frequently. Loss of $\Psi 1290$ also results in correlation changes with m$^1$acp$^3\Psi 1191$, Gm1271, and Cm1639. Together, these results highlight relationships between modifications at different sites and provide testable models for future studies.

## Correlated modification at distant sites on wild-type yeast ribosomes

By ensemble methods, most rRNA positions are almost completely ( > 95%) modified and cluster around functional centers, with a few that are partially modified (*Figure 1—figure supplement 5*; *Birkedal et al., 2015*; *Marchand et al., 2016*; *Taoka et al., 2016*; *Yang et al., 2016*). Taken one position at a time, our model arrives at very similar estimates (*Figure 2—figure supplement 4*). However, ensemble methods cannot detect heterogeneity in the modification pattern of whole molecules. To identify modification heterogeneity in wild-type yeast ribosomes, we searched for subpopulations of rRNA with single molecule modification profiles not expected by chance. Hierarchical clustering of wild type profiles shows no large classes of distinctly modified ribosomes. However, some smaller ( < 10% of total) subpopulations are apparent that have correlated pairs of unmodified nucleotides (*Figure 2E* and *Figure 2—figure supplement 5A*), confirmed by comparing wild type to IVT (*Figure 2F* and *Figure 2—figure supplement 5B*) (see Materials and methods). One pair of significantly correlated positions in 18 S, $\Psi 632$ and $\Psi 766$ (p-value = 1.3e-04, Fisher z-transform test) are guided by the same snoRNA, snR161, likely explaining the basis for this correlation.

We observe a significant correlation between Am100 and Am436 (p-value = 3.1e-04, Fisher z-transform test) as well as between Cm1639 and ac$^4$C1773 in 18 S (p-value = 4.5e-06, Fisher z-transform test) (*Figure 2F*), neither of which share a snoRNA or modification enzyme. Cm1639 (snR70) in the P-site is also correlated with $\Psi 999$ (snR31, E-site, p-value = 1.7e-03, Fisher z-transform test), and ac$^4$C1280 (Kre33, *P*-value = 3.9e-05, Fisher z-transform test) (*Figure 2*). Furthermore, ac$^4$C1773 and m$_2^6$A1782 (Dim1, intersubunit bridge, p-value = 1.7e-04, Fisher z-transform test) are correlated. Some correlated pairs lie close to each other (15–22 Å) in the three-dimensional structure of the mature ribosome (*Figure 2G*), suggesting a structural or functional basis for their coordinate modification status. In 25 S, the three U24-guided positions Cm1437, Am1449, and Gm1450 are all significantly more correlated with each other than expected (p-value = 1.0e-44, Brown's method) (*Figure 2—figure supplement 5*), echoing our observations in the depletion experiment. Several of the significant long-range correlations in wild type show up in other experiments (see below), indicating that concerted modification at those positions is a feature of normal yeast ribosomes.

## Loss of different RNA helicase-related functions result in distinct subpopulations of differently modified rRNA molecules

Previous studies have connected helicase activities required for ribosome biogenesis with changes in 2'O methylation at single positions in ensembles of rRNA molecules (*Martin et al., 2013*). To explore how ATP-dependent RNA helicase functions implicated in ribosome biogenesis and snoRNP dynamics may influence single molecule patterns of rRNA modification, we profiled cells deficient

in two proteins with distinct roles in ribosome biogenesis; Dbp3 (*Weaver et al., 1997*) and Prp43 (*Combs et al., 2006*; *Leeds et al., 2006*). Compared to wild-type control strains at 30 °C or 18 °C, we observe loss of 2′O methylation at specific locations in 18 S and 25 S rRNAs in the Dbp3 knockout strain (*dbp3Δ*) or the cold-sensitive Prp43 Q423N mutant (*prp43-cs*) grown at nonpermissive temperature (*Figure 3A*, *Figure 3—figure supplements 1–3*) consistent with previous ensemble studies (*Figure 2—figure supplement 4C, D*; *Aquino et al., 2021*). Despite the numerous locations at which modification is compromised, hierarchical clustering of 25 S rRNA single-molecule profiles reveals that just 2–3 distinct but related sets of modification profiles describe nearly all ribosomes in both experiments (*Figure 3B*). The triad of 2′O methylations guided by the snoRNA U24 at 25 S positions 1437, 1449, and 1450 surrounding the PET are often left unmodified in a highly concerted manner (*Figure 3B*), as observed in a minority of wild type 25 S rRNA molecules (*Figure 2—figure supplement 5*), and in the snoRNP depletion experiments (*Figure 1*). The pairwise correlations within this triad are significantly higher in both the *dbp3Δ* and the *prp43-cs* mutants relative to wild type (*dbp3Δ* p-value = 3.6e-68, *prp43-cs* 3.5e-11, Brown's method) (*Figure 4* and *Figure 4—figure supplement 1D, E*). To confirm the highly concerted modification of the closely spaced U24-dependent positions, we clustered the underlying raw signal event means from the *dbp3Δ* and *prp43-cs* mutants as above (see Materials and methods) (*Ding et al., 2020*). This reveals two clear subpopulations of reads distinguished by the signal means at positions C1437, A1449, and G1450 in both the *dbp3Δ* and *prp43-cs* mutants (*Figure 3—figure supplement 4*). Therefore, if the U24 snoRNP is unable to guide modification of any of these positions, then all 3 positions are left unmodified. Furthermore, both Dbp3 and Prp43 functions are required for the concerted modification of these nucleotides near the PET.

Prp43 interacts with a number of G-patch proteins that direct it to either the ribosome or the spliceosome (*Chen et al., 2014*; *Heininger et al., 2016*; *Martin et al., 2002*; *Pertschy et al., 2009*; *Tanaka et al., 2007*; *Tsai et al., 2005*). Two of these, Pxr1 and Sqs1, are important for correct pre-rRNA processing (*Banerjee et al., 2015*; *Guglielmi and Werner, 2002*; *Pertschy et al., 2009*). To discover the role of Pxr1 and Sqs1 in rRNA modification, we obtained modification profiles from strains deleted for each. Although deletion of Sqs1 had little effect on modification (see below), loss of Pxr1 produced modification profiles similar to the *prp43-cs* mutant, but with more extreme modification deficiencies (*Figure 3A*). All modifications affected by *prp43-cs* and all but two 18 S 2′O methylations affected by *dbp3Δ* (*Figure 3—figure supplement 1A*) are also affected in *pxr1Δ*. This suggests that loss of Prp43 activity guided by Pxr1, but not that guided by Sqs1, is responsible for the concerted changes in modification pattern observed in the *prp43-cs* strain.

Despite the similarities in modification patterns in the different mutants (*Figure 4* and *Figure 4—figure supplement 1*), they are not identical. For example, in 25 S, positions Am817 and Gm908 (both guided by snR60) and Gm2619 and Um2724 (both guided by snR67) are significantly more correlated in both *pxr1Δ* and *dbp3Δ* relative to wild type (817–908 p-value = 4.7e-17 and 2.4e-33, 2619–2724 p-value = 7.2 e-04 and 3.5e-22, Fisher z-transform tests) (*Figure 4—figure supplement 1D*). However, they are not significantly more correlated in the *prp43-cs* mutant relative to wild type (817–908 p-value = 0.34, 2619–2724 p-value = 0.919, Fisher z-transform test). Pxr1 and Dbp3 both appear to promote concerted modification of positions guided by snR60 and snR67; however, the contribution of Prp43 is less clear. It is possible that the conditional Prp43 mutation is not severe enough to produce a strong block to modification at those sites, or alternatively that Pxr1 has functions in addition to its role in support of Prp43.

Together our data reveal distinct classes of ribosomes that result from a concerted network of modification losses, many of which reside in the functional centers of the ribosome (*Figure 4*). For example, loss of Prp43 and Pxr1 induce a concerted loss of modification of a set of nucleotides in the decoding site of the small subunit (green circles, *Figure 4B*). Loss of Pxr1 leads to concerted loss of a set of modifications in the peptidyl transfer center of the large subunit (blue circles, *Figure 4A*). And all three mutants create a complex set of correlated modification changes in the triad Cm1437, Am1449, and Gm1450 near the protein exit tunnel of the large subunit (purple circles, *Figure 4A*). Concerted modification of this triad is observed in wild type ribosomes (*Figure 2—figure supplement 5*) as well as in the snoRNP depletion experiments (*Figure 1*). As discussed above, a shared snoRNP (e.g. snR60, snR67) may explain part of the concerted modification phenomenon, however in many other cases the mechanisms that underlie concerted modification are not obvious.

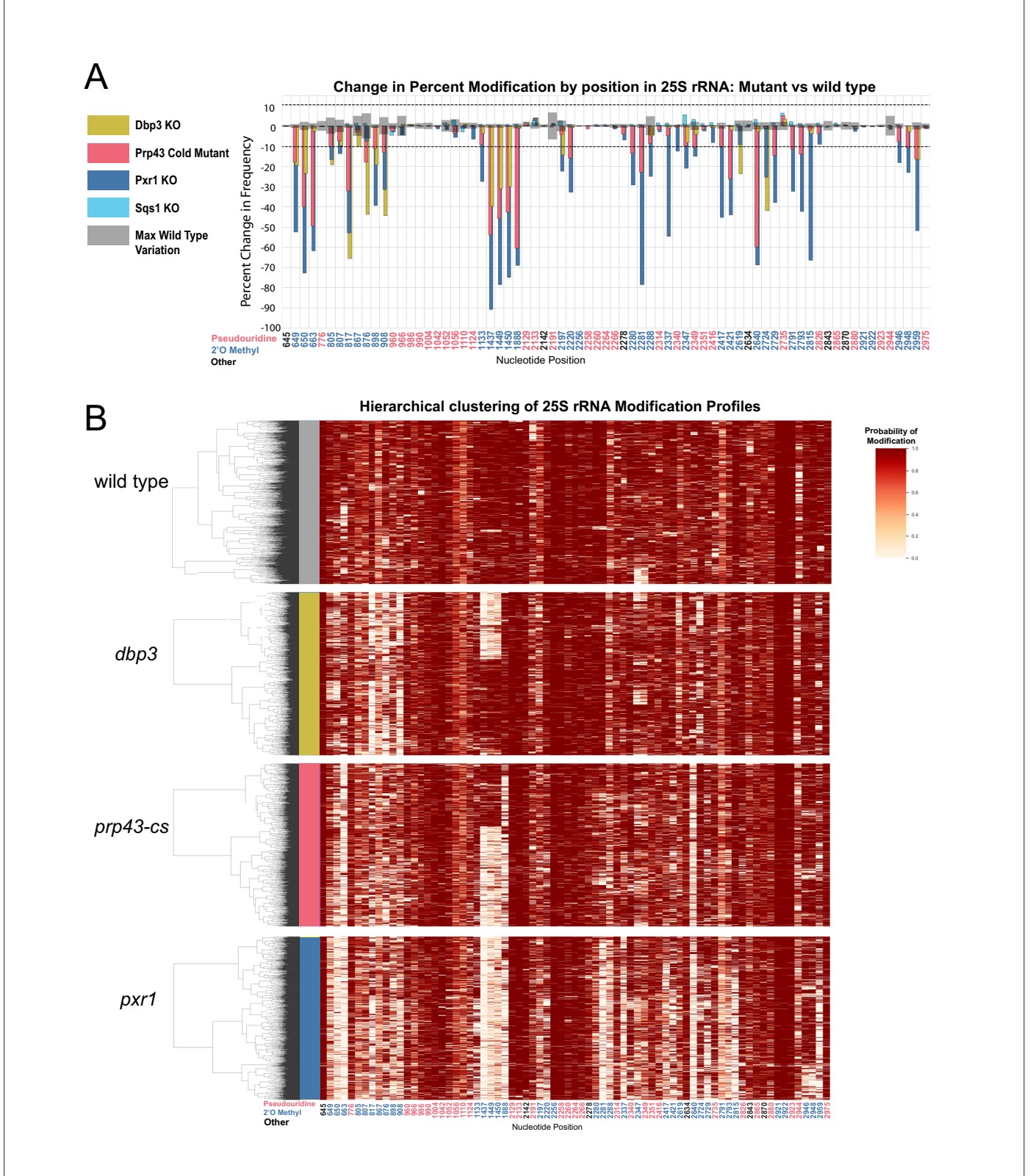

**Figure 3.** Clustering of 25 S rRNA modification profiles and percent change in modification frequency of mutant helicases Dbp3 and Prp43 and G-patch proteins Pxr1 and Sqs1. (**A**) Barplots of the difference between wild-type modification frequency and *dbp3Δ*, *prp43-cs*, *pxr1Δ*, and *sqs1Δ* modification frequencies in 25 S yeast rRNA. Gray bars indicate the variance of wild type rRNA modification at each position and the black dotted lines represent the maximum variance observed at any site. (**B**) Hierarchical clustering of 25 S yeast rRNA modification profiles from wild type, *dbp3Δ*, *prp43-cs*, and

*Figure 3 continued on next page*

Figure 3 continued

*pxr1Δ* (1000 reads in each experiment). Each row is a full-length molecule, each column is a modified nucleotide and the color represents modification probability, see scale.

The online version of this article includes the following figure supplement(s) for figure 3:

**Figure supplement 1.** Clustering of 18 S rRNA modification profiles and percent change in modification frequency upon mutation of helicases Dbp3 and Prp43 and G-patch proteins Pxr1 and Sqs1.

**Figure supplement 2.** Clustering of 18 S rRNA modification profiles and percent change in modification frequency upon mutation of helicases Prp43 and Prp16, compared to wild type controls grown at 30 °C or shifted to 18 °C for 1 hr.

**Figure supplement 3.** Clustering of 25 S rRNA modification profiles and percent change in modification frequency upon mutation of helicases Prp43 and Prp16, compared to wild-type controls grown at 30 °C or shifted to 18 °C for 1 hour.

**Figure supplement 4.** Clustering of underlying events to search for patterns of modification in the Dbp3 KO and Prp43 cold mutant.

## Resilience of rRNA modification profiles to acute splicing perturbations and environmental treatments

Ribosome biogenesis and splicing are connected processes in yeast. For example, Prp43 also mediates disassembly of spliceosomes (*Combs et al., 2006*; *Leeds et al., 2006*; *Martin et al., 2002*) aided by the G-patch protein Spp382 (also called Ntr1) (*Christian et al., 2014*; *Fourmann et al., 2016*; *Pandit et al., 2006*; *Tanaka et al., 2007*; *Tsai et al., 2005*). In addition, a number of snoRNAs are encoded within introns of genes important in ribosome biogenesis and translation, and their synthesis can be compromised by mutations that affect splicing (*Ooi et al., 1998*; *Petfalski et al., 1998*; *Piekna-Przybylska et al., 2007*; *Vincenti et al., 2007*). To determine if loss of splicing functions affect rRNA modification, we acquired single molecule modification profiles for ribosomes from an Spp382/Ntr1 mutant, a cold sensitive mutant of Prp16 (*prp16-302*) (*Madhani and Guthrie, 1994*; *Tseng et al., 2011*) that accumulates splicing intermediates, and a deletion of Dbr1 (*Chapman and Boeke, 1991*) that prevents debranching of the intron lariat, a reaction that promotes processing of some intronic snoRNAs, in particular U24 (*Ooi et al., 1998*).

Using the threshold of >10% change in modification relative to wild type established above (*Figure 1—figure supplement 4* and *Figure 3A*), we examined splicing-related perturbations for effects on rRNA modification that might be mediated through loss of one or more intronic snoRNAs (*Figure 5*). We observe a 36.8% reduction in modification for 18 S Ψ106 (guided by snR44 from intron 2 of *RPS22B*) and an 11.0% reduction in modification frequency for 18 S Am974 (guided by snR54 from intron 1 of *IMD4*) as a consequence of the loss of Dbr1. There are hints that other intron-encoded snoRNAs may be affected, for example U24, however none of these reached our conservative threshold (*Figure 5*). Neither the *prp16-302* nor *spp382-1* mutant produced a modification defect that passed the threshold (*Figure 5*), suggesting that deficiencies in ongoing splicing do not create noticeable changes in rRNA modification pattern. Both the *prp16-302* and the *spp382-1* mutant alleles are viable hypomorphs of essential genes, and might not be sufficiently severe to trigger steady state loss of snoRNAs or other perturbations we could detect. In the case of the Dbr1 null strain, there are alternative snoRNA maturation pathways that are independent of splicing, for example via Rnt1 (*Villa et al., 1998*), and incompletely processed U24 still can guide modifications at its corresponding locations (*Aquino et al., 2021*; *Ooi et al., 1998*; *Petfalski et al., 1998*). These results argue that the modification defects observed for prp43-cs arise primarily from its deficiencies in promoting ribosome biogenesis, and not from the loss of its contributions to splicing.

In nature, yeast cells adapt to different environments by changing patterns of gene expression (*Gasch et al., 2000*). With a few exceptions (*Liu et al., 2021b*), how changes in rRNA modification contribute to stress adaptation in yeast is still largely unknown. To test whether single-molecule modification profiles were altered by various, rapidly changing environmental conditions known to affect ribosome function and biogenesis (*Gasch et al., 2000*; *Warner, 1999*), we isolated and acquired profiles of rRNA isolated from cells at stationary phase (*Ju and Warner, 1994*; *Talkish et al., 2016*), after a 1 hr shift to potassium acetate to induce starvation, treated with rapamycin (TOR kinase inhibitor) for 1 hr to block nutrient signaling (*Cardenas et al., 1999*; *Hardwick et al., 1999*; *Heitman et al., 1991*; *Honma et al., 2006*; *Powers and Walter, 1999*; *Talkish et al., 2012*; *Vanrobays et al., 2008*), treated with cycloheximide to block translational elongation (*Obrig et al., 1971*) and create ribosome

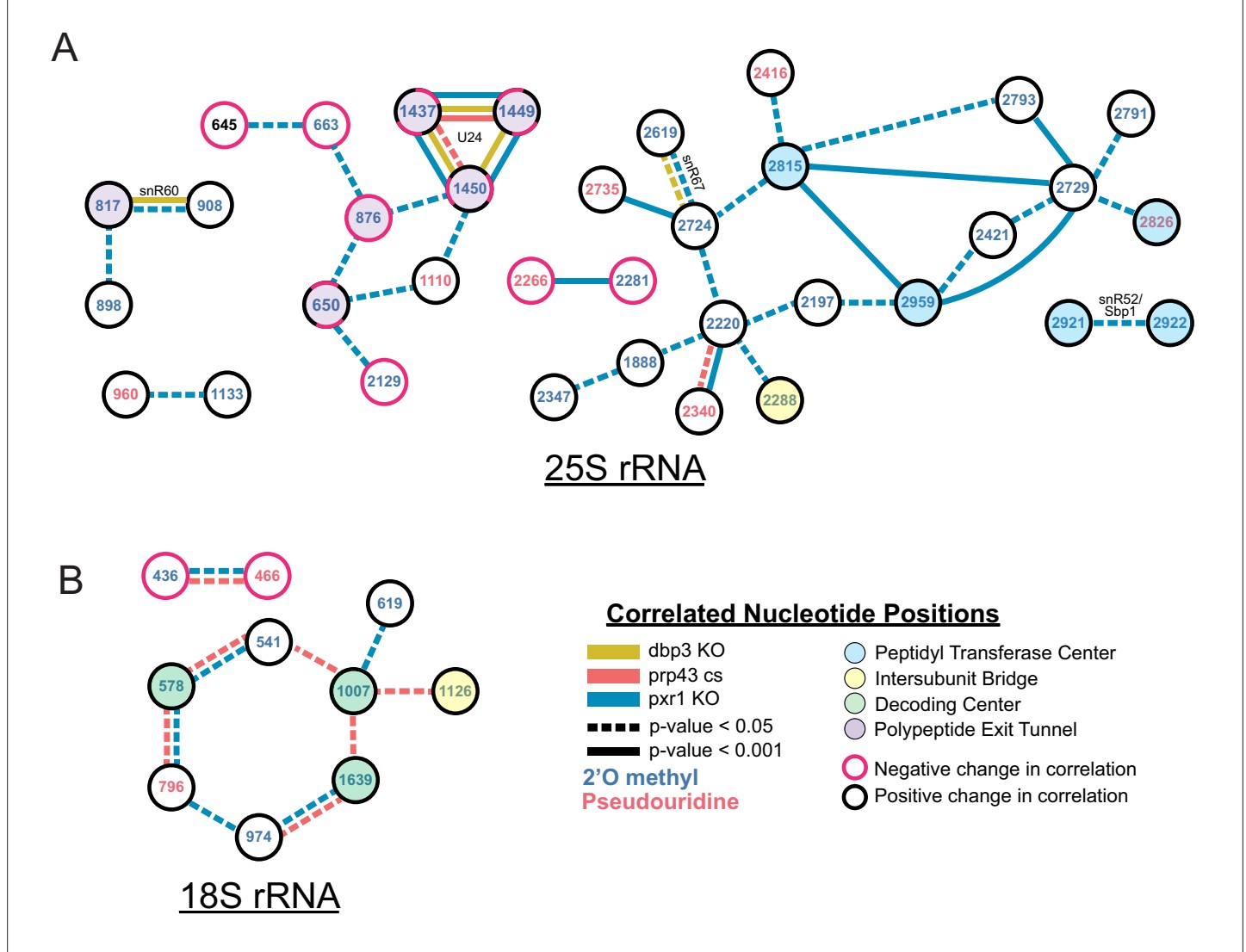

**Figure 4.** Changes in correlated nucleotide positions in *dbp3Δ*, *prp43-cs*, or *pxr1Δ* mutants. Graphical representation of modification correlation changes, significance, and location within the functional centers of the ribosome. Pairs of correlated nucleotide changes (nodes) are shown for each mutant (colored edges) relative to wild type yeast 25 S rRNA (**A**) and 18 S rRNA (**B**). In cases where correlated pairs show differential changes in correlation in different mutants (eg. U24 modifications), node color rings are fragmented with the appropriate mutant edge connecting to either the magenta (negative change in correlation relative to wild type) or black (positive change in correlation relative to wild type) portion of the node.

The online version of this article includes the following figure supplement(s) for figure 4:

**Figure supplement 1.** Correlation analysis of *dbp3Δ*, *prp43-cs*, and *pxr1Δ*.

collisions (*Simms et al., 2017*), or after cold shock. In none of these treatments did we detect substantial changes in modification profile (*Figure 5* and *Supplementary file 1A,B*).

Together, these observations indicate that most annotated modification patterns on rRNA are refractory to rapid alterations by dramatic changes in the physiological conditions we tested. In the case of environmental treatments for 1 hr, cell numbers increased by approximately 1.2–2.4 fold, providing opportunity for new, differently modified ribosomes to be synthesized and detected in our experiments. The conditions we tested are not known to trigger loss of snoRNPs, therefore any rapid changes in modification that might have been observable would have had to rely on either repressing snoRNP function or activation of enzymes that reverse the modification at pre-existing modified sites. At this time, there are no known mechanisms for repressing snoRNP function or enzymes that would reverse either pseudouridylation or 2´O methylation of ribose in RNA. However, these results do not

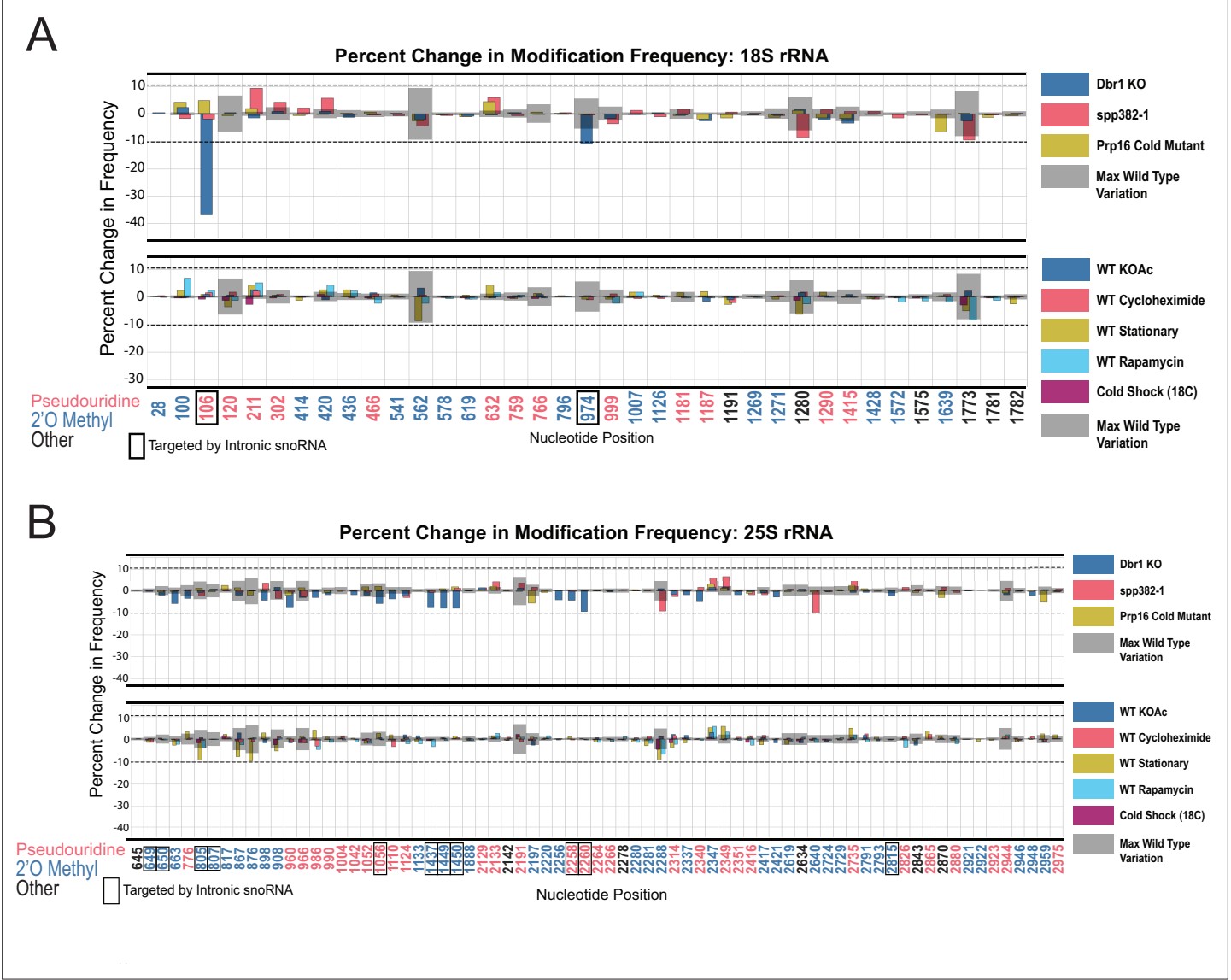

**Figure 5.** Resilience of yeast rRNA modifications to a variety of splicing mutants and experimental conditions. Barplots of the difference between wild-type modification frequency and *dbr1Δ*, *spp382-1*, *prp16-302*, KOAc treated, cycloheximide treated, stationary, rapamycin treated and cold shock yeast modification frequencies in yeast 18 S (**A**) and 25 S (**B**) rRNA. Gray bars indicate the variance of wild-type rRNA modification at each position and the black dotted lines represent the maximum variance across sites.

exclude the possibility that chronic genetic or environmental conditions might alter rRNA modification and ribosome function over the long term.

## Discussion

To discover relationships and dependencies between distant modifications in rRNA, we captured single molecule modification profiles of *S. cerevisiae* 18 S and 25 S rRNA. By depletion of the two main classes of snoRNPs responsible for the bulk of rRNA pseudouridylation and 2′O methylation, we validated the framework for our method and found that these two classes of snoRNP-dependent modification are largely independent of each other (*Figure 1*). We resolved populations of ribosomes that differ by a single modification, and identified instances of concerted modification of sets of nucleotides in the wild type ribosome population (*Figure 2*). Analysis of the single molecule modification profiles resulting from loss of two distinct helicase activities provided by Dbp3 and Prp43 (and its G-patch protein Pxr1) reveals a complex set of long-range concerted effects on modification, with

implications for ribosome biogenesis and function (*Figures 3, 4 and 6*). Finally, we found that most annotated modifications are refractory to changes in physiological conditions, inhibitors of ribosome function, or stress on splicing, which is critical to ongoing ribosome biogenesis (*Figure 5*; *Warner, 1999*). These results provide new perspectives on ribosome heterogeneity as represented by RNA modification patterns and open a path to single molecule analysis of RNA modification for other RNAs.

## Concerted modification in the polypeptide exit tunnel

Distinct subsets of 25 S rRNAs collectively missing 2′O-methyl modification at positions Cm1437, Am1449, and Gm1450 accumulate together and form a major subset of ribosomes in Pxr1, Prp43, and Dbp3 mutants (*Figures 3 and 4*). Importantly, wild type ribosomes show a similar pattern of concerted modification at these residues (*Figure 2—figure supplement 5*), suggesting that these are not exclusively a result of the mutations. The polypeptide exit tunnel (PET), acquires modification in a concerted fashion (*Figures 3 and 4*, *Figure 2—figure supplement 5*). These three nucleotides line the PET, where interactions with the nascent polypeptide chain can influence protein folding (*Choi et al., 2018*), within a few angstroms of conserved loops of ribosomal proteins uL4 (rpL4) and uL22 (rpL17) (*Ben-Shem et al., 2011*). These protein loops insert into the PET to form the constriction site and may act as an 'exit gate' (*Figure 6A and B*; *Nakatogawa and Ito, 2002*; *Wilson et al., 2020*; *Zhang et al., 2013*). Furthermore, these three positions are in domain 0 of ribosomal rRNA, which acts as a central hub around which the other six 25 S rRNA domains fold (*Klinge and Woolford, 2019*; *Petrov et al., 2014*). In the absence of Cm1437, Am1449, and Gm1450, the rRNA and the loops of uL4 and uL22 may not be properly positioned, affecting the structure and chemistry of the PET, translation, and protein folding. Future work will be necessary to assess the functional differences between the two populations of ribosomes we observe that differ by concerted modification of these PET nucleotides.

## Distinct and overlapping helicase functions during ribosomes biogenesis

At least 21 putative RNA helicases drive yeast ribosome biogenesis forward by enabling pre-rRNA rearrangements and releasing snoRNPs (*Martin et al., 2013*). Recent work (*Aquino et al., 2021*) supports a model in which Dbp3 and Prp43 release certain snoRNPs to allow subsequent modification of adjacent sites by a second snoRNP. Here, we identify concerted changes in modifications over much longer distances in rRNA when the activity of Dbp3 or Prp43 is compromised, suggesting the regulation of these helicases is not only important for locally overlapping modified positions, but also for modifications separated by large stretches of sequence (*Figure 3*, *Figure 3—figure supplements 1–3*). Furthermore, our work shows that Pxr1, but not Sqs1, is the main G-patch protein important for Prp43 function during rRNA modification.

In the absence of ribosomal helicase-related functions (Prp43, Pxr1, and Dbp3) we observe a large set of overlapping but not identical changes in modified positions. Single molecule profiling reveals distinct hubs of concerted modifications, many of which reside in functional centers of the ribosome (*Figures 3 and 4* and *Figure 3—figure supplement 1*). In the case of 18 S rRNA, a hub of nucleotides are left unmodified in a concerted fashion when Prp43 and Pxr1 activity are compromised, creating a distinct subset of undermodified subunits. Previous work (*Aquino et al., 2021*; *Bohnsack et al., 2009*) showed that when Prp43 is absent from pre-ribosomes, the snR55 snoRNP is retained on pre-ribosomes but its target site 18 S rRNA Um1269 remains modified. This suggests snR55 can bind and direct modification, but then is not efficiently released from the pre-ribosome. Consistent with this, the Prp43-Q423N protein is more tightly associated with snR55 and other snoRNAs than wild type Prp43 (*Leeds et al., 2006*).

Here we show that the Prp43-Q423N mutant protein permits modification of Um1269 by snR55, but causes concerted loss of modification at nucleotides that surround the Prp43 binding site in the base of helix 44, which appear to be secondary to snR55 modification (*Figure 6C*; *Bohnsack et al., 2009*). Detectable snR55 is associated with pre-ribosomes in the absence of Dbp3, but its ability to modify U1269 is blocked (*Figure 3—figure supplement 1A*; *Aquino et al., 2021*). Unexpectedly, in the absence of Dpb3 where snR55-dependent modification does not occur, these secondary modifications are observed (*Figure 3—figure supplement 1*; *Aquino et al., 2021*). Together, this suggests a

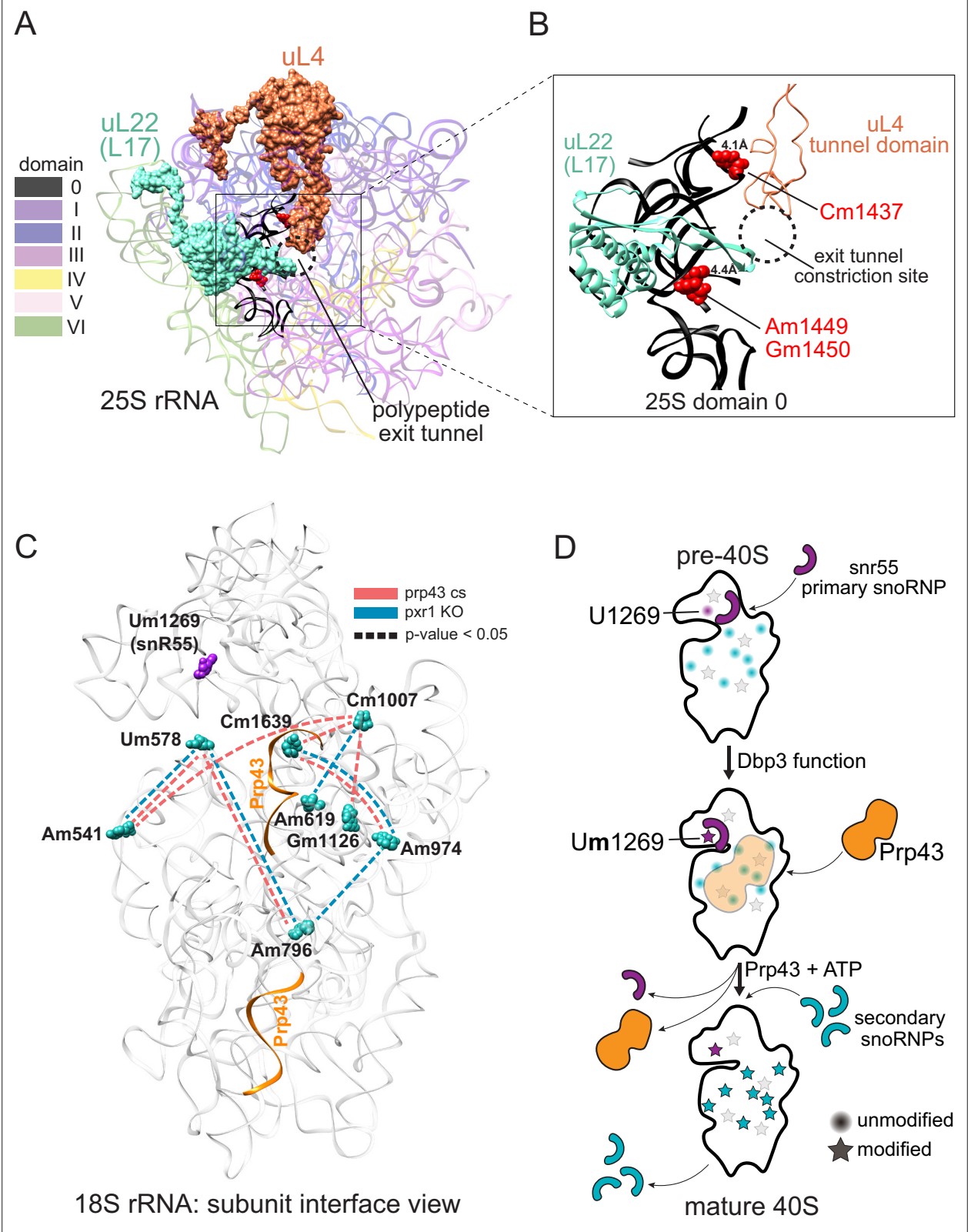

**Figure 6.** Models describing the functional basis for concerted modification at specific locations in the ribosome. (**A**) Crystal structure model of yeast 25 S rRNA and ribosomal proteins uL4 and uL22 in surface view (PDB:4V88; *Ben-Shem et al., 2011*). rRNA domains are color coded according to the RiboVision Suite (*Bernier et al., 2014*). The distal end of the polypeptide exit tunnel is indicated. U24-guided modified nucleotides Cm1437, Am1449, and Gm1450 are shown in blue. (**B**) Focused view of the uL4 tunnel domain and the internal loop of uL22 forming the exit tunnel constriction sites. 25 S

*Figure 6 continued on next page*

*Figure 6 continued*
rRNA domain 0 is shown in black. 4.1Å and 4.4Å represent the distance between the indicated, modified nucleotide and the internal loops of uL4 and uL22 respectively. (**C**) Crystal structure model of yeast 18 S rRNA showing changes in concerted loss of modifications in *prp43-cs*, or *pxr1Δ* mutants when snR55 (Um1269, purple) is retained on the pre-ribosome in the absence of Prp43 (*Bohnsack et al., 2009*). Prp43 crosslinking sites are indicated in orange. (**D**) Model showing the hierarchical function of Dbp3 and Prp43, to promote concerted modification of 18 S rRNA nucleotides, during ribosome biogenesis. Circles and stars represent unmodified and modified positions, respectively. Modified nucleotides in the 5′ end of 18 S rRNA, proposed to occur early and co-transcriptionally, are represented as light gray stars. Color scheme is the same as panel C.

hierarchical model for the action of Dbp3 and Prp43/Pxr1 during co-transcriptional ribosome biogenesis (*Figure 6D*).

The Prp43-Q423N mutant also leads to undermodification of other nucleotides nearby in three-dimensional space, but their loss is not concerted with the others on a single molecule basis. These nucleotides are in the 5′ end of 18 S rRNA, and likely to be modified co-transcriptionally (*Kos and Tollervey, 2010*) prior to snR55 binding. We propose that modifications at 5′ end of 18 S occur early and co-transcriptionally, independently of later modifications. As transcription and ribosome assembly progresses, Dbp3 becomes necessary for methylation of U1269 by snR55 in 18 S, possibly by promoting a conformational change in the pre-ribosome or snR55. The absence of Dbp3 does not prevent snR55 binding to the pre-ribosome in a fashion that permits modification at positions surrounding the Prp43 binding site (*Figure 3—figure supplement 1*). However, after snR55 modification of U1269, Prp43 activity is required for snR55 release, allowing other snoRNPs to modify positions around the Prp43 binding site. Because the Prp43-Q423N protein is slow to dissociate, snR55 and mutant Prp43 are retained on the pre-ribosome (*Bohnsack et al., 2009*; *Leeds et al., 2006*), occluding the other snoRNPs, resulting in concerted loss of those modifications (*Figure 6C and D*).

Prp43 is present in multiple, consecutive pre-ribosomal particles and is thought to bind to different positions along the pre-rRNA (*Bohnsack et al., 2009*; *Lebaron et al., 2005*), providing opportunity for hierarchical relationships in modification at other locations. During its function in the termination of pre-mRNA splicing, Prp43 is proposed to bind the 3′ end of U6 snRNA and translocate along it, resulting in removal of U6 from the spliceosome (*He et al., 2017*; *Toroney et al., 2019*). Because many biochemical principles are shared between the spliceosome and the ribosome (*Staley and Woolford, 2009*), we propose that Prp43 acts in an analogous manner to release certain snoRNPs from the pre-ribosome upon completion of modification. Since other modification sites in 25 S rRNA are affected by loss of Prp43 activity, it seems likely that Prp43 binding at multiple locations promotes release of snoRNPs from the pre-ribosome. Thus, hubs of concerted loss of modification in the Prp43 mutant might reflect critical points in ribosome biogenesis where Prp43 removes key snoRNAs.

## Is U24 snoRNP-dependent methylation distributive or processive?

The exact mechanism by which snoRNPs bind to and are released from the pre-ribosomal particles during assembly is still unknown. Models must account for the fact that many snoRNP binding sites overlap in both the linear rRNA sequence and the three-dimensional spaces of the assembling ribosome. In this work, we document concerted modification at several sets of locations, some distant on the sequence but near in space in the mature ribosome, and others that may or may not share a snoRNA guide (*Figures 1–4*, *Figure 1—figure supplement 3*, *Figure 2—figure supplement 5*, and *Figure 4—figure supplement 1*). The most striking example of concerted modifications are the U24-guided modifications at Cm1437, Am1449, and Gm1450 in the PET. The depletion experiments (*Figure 1*) provide clear evidence that most ribosomes are either fully modified or completely unmodified at those three sites, and the loss of modification in the Dbp3 and Prp43 mutants for this group is similar (*Figure 3* and *Figure 4—figure supplement 1*).

Does one U24 binding event lead to all three modifications? Or does each modification event require an independent binding event? In yeast, U24 and three other C/D box snoRNPs (U18, snR13, and snR48) are thought to modify 2–3 nucleotides that reside next to each other in sequence. It has been suggested that alternatively folded forms of the snoRNA within these snoRNPs may guide the individual modifications (*van Nues and Watkins, 2017*). This proposal and our observations open the possibility of a processive mechanism for concerted local modification events in which one snoRNP binding event with snoRNA refolding in between explains concerted sequential modification dynamics. A distributive mechanism would require independent, stochastic snoRNP binding events, as expected

for snoRNPs that guide modification of distant sites, or where two different snoRNPs guide closely spaced modified sites. The latter case should produce a set of rRNAs with partial modifications in different combinations at these positions, and this is not observed (*Figures 1 and 3*, *Figure 2—figure supplement 5*, and *Figure 4—figure supplement 1*), lending weight to the idea that closely spaced modifications guided by a single snoRNP may occur processively.

The diameter of a snoRNP may approach 100 Å (*Duan et al., 2009*; *Lin et al., 2011*), whereas the yeast ribosome is approximately 250 Å in diameter. Thus, if two different snoRNPs have overlapping binding sites that guide two closely spaced modifications, they almost certainly must act sequentially. In this instance, concerted loss of modification could occur if the pre-rRNA substrate fails to adopt a conformation that renders the local region containing both sites accessible. However, this may be inadequate to explain concerted modification of closely spaced nucleotides by the same snoRNP, such as the U24-guided modifications at Cm1437, Am1449, and Gm1450 (*Figure 3* and *Figure 2—figure supplement 5*). In this case if the site is inaccessible, all three positions would be unmodified. However, if the site is accessible and modification proceeds distributively then intermediates representing partial modification should be observed, in particular under depletion of C/D box snoRNPs (*Figure 1*) or loss of Dbp3 (*Figure 3*; *Aquino et al., 2021*). Instead, two main populations of ribosomes are observed, all modified or all unmodified at these locations, opening the possibility that modification of these sites by U24 snoRNP is highly processive. In this view, when the site is accessible all three sites become methylated without dissociation of the U24 snoRNP from the pre-ribosome. In Dbp3 mutants and wild-type cells, most U24 is associated with the pre-ribosome (*Aquino et al., 2021*), suggesting that either (1) there are three high-affinity binding sites at which release and rebinding of U24 occurs distributively at a rate that prevents detection of partially modified intermediates, or (2) there is a single binding site at which the target nucleotides are bound and methylated processively after a single U24 snoRNP-binding event.

## Prospects and limitations

We developed a hidden Markov model-based approach that allows (1) single-molecule profiling and clustering of RNAs to visualize high-level relationships between features in a population of complete molecules, (2) the ability to test for changes in correlations between any given pair of modifications on the same molecule, and (3) a way to estimate the fraction of modification of each site. We used wild-type rRNA as our fully modified training example, with the clear understanding that not all wild-type molecules are fully modified (*Taoka et al., 2016*). In several instances, we confirmed that this assumption had little effect on performance of the model (*Figure 1—figure supplement 2*). Positions where the signal-to-noise ratio between unmodified and modified nucleotides is small are less accurately predicted compared to other positions. A sense of this limitation can be observed as occasional miscalling of positions as modified in the in vitro transcripts (red positions in the IVT heatmap of *Figure 1A*; per position accuracy is shown in *Figure 1—figure supplement 2E*). This limitation is a function of the sequence context with which the modification occurs and vagaries of nanopore electronic signal variation. Another limitation arises when the training samples do not have enough information to resolve dense clusters of modifications. Where this was a concern, we validated patterns of modification by clustering the raw signal means (*Figure 1—figure supplement 4* and *Figure 3—figure supplement 4*). While there is some evidence that unknown modified kmer distributions can be estimated using known kmer distributions (*Ding et al., 2021*), generating more specific modification training data sets that contain all combinations of partially modified closely spaced clusters of nucleotides may be required to produce more accurate and general modification detection algorithms. This is especially true if de novo detection of modifications within complex sequences is the goal (*Leger et al., 2019*; *Stoiber et al., 2016*).

While our analysis focuses on mature, full-length rRNA, the capture of and profiling of pre-rRNA intermediates, nascent chromatin-associated pre-rRNA, and rRNA turnover products will certainly further resolve and enhance our understanding of how the concerted placement and timing of rRNA modifications occurs during the lifetime of the ribosome. Recently, alterations of rRNA modification have been detected in human diseases and cancer (*Babaian et al., 2020*; *Barros-Silva et al., 2021*; *Janin et al., 2020*), suggesting that monitoring of concerted changes in rRNA modification might serve as important biomarkers for human disease. Importantly, the model training paradigm we have developed to profile modifications can easily be applied to other nucleic acids of interest. Recent

advancements in nanopore sequencing of full-length human rRNAs (*Jain et al., 2021*) and *E. coli* tRNAs (*Thomas et al., 2021*) make these heavily modified molecules ripe for single-molecule profiling to understand how RNA modifications regulate translation (*Erales et al., 2017*; *Krogh et al., 2016*; *Schimmel, 2018*). Furthermore, it is becoming increasingly clear that modifications of pre-mRNAs and the snRNAs within the splicing machinery are important for regulating alternative splicing and misregulation of modifications are associated with disease (*Bartosovic et al., 2017*; *Lin et al., 2016*; *Zheng et al., 2013*). Thus, the ability to monitor the modification status at each position across an mRNA molecule, along with its alternative splicing pattern (*Drexler et al., 2020*; *Tang et al., 2020*), will become increasingly important for basic biological understanding and discovery.

# Materials and methods

## Key resources table

| Reagent type (species) or resource | Designation | Source or reference | Identifiers | Additional information |
|---|---|---|---|---|
| Strain, strain background (*Saccharomyces cerevisiae*) | *Saccharomyces cerevisiae*, various strains and genetic backgrounds | | | See *Supplementary file 1D* |
| Strain, strain background (*Escherichia coli*) | XL1 Blue | In-house | | Electrocompetent cells |
| Recombinant DNA reagent | T7-18S | This paper | | HindIII digested, run-off in vitro transcription (see Materials and methods) |
| Recombinant DNA reagent | T7-25S | This paper | | HindIII digested, run-off in vitro transcription (see Materials and methods) |
| Sequence-based reagent | sequencing adapter | Integrated DNA Technologies | Sequencing adapter – *Supplementary file 1E* | /PHOS/GGCTTCTTCTTGCTCTTAGGTAGT AGGTTC |
| Sequence-based reagent | s.c. 18 S splint | Integrated DNA Technologies | Sequencing adapter splint – *Supplementary file 1E* | CCTAAGAGCAAGAAGAAGCCTAATGATC CTTCC |
| Sequence-based reagent | s.c. 25 S splint | Integrated DNA Technologies | Sequencing adapter splint – *Supplementary file 1E* | CCTAAGAGCAAGAAGAAGCCACAAATCA GACAA |
| Sequence-based reagent | s.c. IVT 18 S splint | Integrated DNA Technologies | Sequencing adapter splint – *Supplementary file 1E* | CCTAAGAGCAAGAAGAAGCCAGCTTTAA TGATC |
| Sequence-based reagent | s.c. IVT 25 S splint | Integrated DNA Technologies | Sequencing adapter splint – *Supplementary file 1E* | CCTAAGAGCAAGAAGAAGCCAGCTTACA AATCA |
| Commercial assay or kit | Gibson Assembly Master Mix | New England Biolabs | E2611L | |
| Commercial assay or kit | MEGAscript T7 transcription kit | Invitrogen | AM1334 | |
| Commercial assay or kit | MinION Mk1B Sequencing Device | Oxford Nanopore Technologies | MIN-101B | |
| Commercial assay or kit | Flongle Adapter | Oxford Nanopore Technologies | ADP-FLG001 | |
| Commercial assay or kit | Direct RNA Sequencing Kit | Oxford Nanopore Technologies | SQK-RNA002 | |
| Commercial assay or kit | Flow Cell (R9.4.1) | Oxford Nanopore Technologies | FLO-MIN106D | |
| Commercial assay or kit | Flongle Flow Cell (R9.4.1) | Oxford Nanopore Technologies | FLO-FLG001 | |
| Commercial assay or kit | Flow Cell Priming Kit | Oxford Nanopore Technologies | EXP-FLP002 | |
| Commercial assay or kit | Flongle Sequencing Expansion | Oxford Nanopore Technologies | EXP-FSE001 | |
| Chemical compound, drug | Rapamycin | Research Products International | R64500-0.001 | |
| Chemical compound, drug | Potassium acetate | EMD Millipore | PX1330-1 | |
| Chemical compound, drug | Cycloheximide | Sigma | C7698-1G | |

## Growth of yeast strains

Yeast strains *GAL-NOP58* and *GAL-CBF5* are described in *Lafontaine et al., 1998*; *Lafontaine and Tollervey, 1999*. Cells were grown at 30 °C in YEPgal liquid medium (2% galactose, 2% peptone, 1% yeast extract) or shifted to liquid YEPD (2% dextrose, 2% peptone, 1% yeast extract) to mid-log phase (OD$_{600}$ = 0.25–0.5) for 16 hr to repress expression of Nop58 or Cbf5. Cells were harvested by centrifugation and RNA was isolated. Unless indicated, all other strains were grown in YEPD at 30 °C to mid-log phase. Cells exposed to various environmental conditions were treated as follows: 1% KOAc (1 hr, 30 °C), cycloheximide (1 μg/ml for 1 hr), or rapamycin (200 ng/ml for 1 or 5 hr). Stationary phase cells were grown to an OD$_{600}$ = 10. Strains carrying *prp16-302* (*Madhani and Guthrie, 1994*) and *prp43-Q423N* (*Leeds et al., 2006*) mutations, and wild type (*Schattner et al., 2004*) were grown to mid log phase at 30 °C and shifted to 18 °C for 1 hr by addition of an equal volume of 6 °C YEPD. The *spp382-1* strain is described in *Pandit et al., 2006*. The *pxr1Δ* strain is described in *Banerjee et al., 2015*. The strains deleted for the *SNR80* (YWD448a), *SNR83* (YWD451a), or *SNR87* (YWD452a) genes are described in *Schattner et al., 2004*. Yeast strains deleted for the *SNR4* and *SNR45* genes are described in *Parker et al., 2018*. The *dbr1Δ*, *dbp3Δ*, and *sqs1Δ* deletion strains were obtained from Open BioSystems. All yeast strains and genotypes can be found in (*Supplementary file 1D*).

## RNA isolation

RNA was extracted from approximately five total OD$_{600}$ of cells (usually 10 ml culture at OD$_{600}$ = 0.5 for mid log cells, 0.5 ml of stationary cells at OD$_{600}$ = 10) using a hot phenol protocol 1 described in *Ares, 2012*.

## In vitro synthesis of 18S and 25S rRNA

Unmodified yeast 18 S and 25 S rRNAs were transcribed in vitro from plasmids encoding T7-18S and T7-25S sequences using T7 RNA polymerase. PCR products encoding 18 S and 25 S rDNA were amplified from the plasmid pWL155 which contains the *RDN1-1* gene fused with the *GAL* promoter at the 5′ end (*Liang and Fournier, 1997* a kind gift from Jelena Jakovlievic) and cloned into a T7 promoter-containing plasmid digested with EcoRI and HindIII using Gibson Assembly (NEB). The resulting plasmids were then digested with HindIII and run-off transcription was performed using the MEGAscript T7 kit (Invitrogen) following the manufacturer's instructions. T7-18S and –25 S in vitro transcription reactions were evaluated by gel electrophoresis for bands of correct size that correspond to 18 S and 25 S rRNAs. Transcription reactions were extracted and purified with phenol:chloroform:isoamyl alcohol (25:24:1), ethanol precipitated and resuspended in nuclease-free H2O. Purified T7-18S and –25 S rRNA transcripts were then quantified on a NanoDrop spectrophotometer and pooled in equimolar ratios for sequencing library preparation. The T7 run-off transcription reactions terminate in a 3′ end generated by HindIII digestion and thus include an additional AAGCU sequence not present in endogenous 18 S and 25 S rRNAs. Therefore, T7-18S and T7-25S splint oligonucleotides were used to capture the 3′ end of T7 transcribed rRNAs (see below, *Supplementary file 1E*).

## Sequencing library preparation

Direct RNA sequencing libraries were constructed using the SQK-RNA002 (Oxford Nanopore Technologies) kit following the manufacturer's protocol with the following modifications. Briefly, 750 ng of total yeast RNA was used as input material. To facilitate ligation of sequencing adapters to endogenous yeast 18 S and 25 S rRNA, 1 μl of 10 pmol/μl custom oligonucleotide duplexes complementary to the 3′ ends of 18 S and 25 S rRNA and the 5′ end of the ONT RMX sequencing adapter were used instead of the kit provided RTA adapter (*Supplementary file 1E*). To create duplexes, 100 pmol of either 18 S or 25 S splint oligo was incubated with 100 pmol of sequencing adapter and nuclease free H20 in a total volume of 10 μl. Reactions were heated to 95 °C for 2 min and gradually cooled at 65 °C for 10 min, 48 °C for 10 min, room temperature for 10 min and then placed on ice. Annealed oligonucleotide duplexes targeting 18 S and 25 S rRNAs were then pooled in equimolar ratio and 1 μl of the pool was used for sequencing library preparation. In the case of T7 rRNA sequencing libraries, T7-18S splint and T7-25S splint oligos were used to capture the 3′ end generated by HindIII digestion and run-off transcription. To enhance ligation efficiency during library preparation, the first and second ligation steps were increased from 10 min to 15 min and performed at room temperature. Reverse transcription was omitted. Sequencing-adapted libraries were eluted in 21 μl of elution buffer.

## Nanopore sequencing

RNAs extracted from *GAL-NOP58* and *GAL-CBF5* strains, and in vitro transcribed RNA were sequenced on the MinION Mk1B sequencer using MinION FLO-MIN106D R9.4.1 flow cells (Oxford Nanopore Technologies) following the manufacturer's instructions. Twenty μl of Sequencing libraries was mixed with 17.5 μl of $H_2O$ and 37.5 μl of RRB buffer. A total of 75 μl of the prepared sequencing library was loaded onto a flushed and primed flow cell and sequenced for 12–48 hr depending on the lifetime of active pores. RNAs extracted from all other strains and growth conditions were sequenced on the MinION Mk1B sequencer using Flongle FLO-FLG001 R9.4.1 flow cells. Flongle flow cells were flushed and primed with 120 μl of flush buffer mix (117 ul FLB and 3 ul FLT). Thirty μl of prepared sequencing library (described above) was loaded onto the flow cell and sequenced for 8–24 hr. Sequencing experiments were controlled using the MinKNOW software (Oxford Nanopore Technologies). Technical replicates, where libraries prepared from the same RNA sample but run on different flow cells, and biological replicates, where the full experiment was repeated, can be found in (**Supplementary file 1F**).

## Data preprocessing

Basecalling was done using the RNA model from Guppy v3.1.5 + 781ed57. To analyze specific subsets of reads more efficiently, we split the multi-fast5 reads into individual reads using the `multi_to_single_fast5` command from https://github.com/nanoporetech/ont_fast5_api. We then created an index file matching a fast5 to a fastq entry using `nanopolish index` from https://github.com/jts/nanopolish (**Simpson et al., 2017**). The reference sequence for the *S. cerevisiae* 18 S and 25 S rRNA came from **Engel et al., 2014**. Initial basecalled sequence to reference alignment was done via minimap2 version 2.17-r943-dirty from https://github.com/lh3/minimap2 using the --MD flag which speeds up processing of signalAlign (**Li, 2018**). Alignment files were sorted and filtered using samtools version 1.9 by flag `-F 2308` which filters out unmapped reads, non-primary alignment reads and supplementary alignment reads (**Li et al., 2009**). Given that nanopore sequencing with RNA is 3′–5′, in order to filter for 'full length' reads we used `samtools view` to select for reads that covered the first 15 bases of either 18 S or 25 S rRNAs (**Li et al., 2009**). Read information and quality control metrics in (**Supplementary file 1F**) were gathered using pycoQC version v2.5.0.23 (**Leger and Leonardi, 2019**).

## SignalAlign pipeline

### Model definition

We initialized the transition probabilities from previous signalAlign r9.4 models. The initialized kmer distributions were defined in `r9.4_180 mv_70bps_5mer_RNA` from ONT https://github.com/nanoporetech/kmer_models. Unlike previous kmer model modification detection algorithms, we chose to model modifications independently from other modifications of the same class in order to maintain the same informational inputs to each modification position. So, we iteratively redefined shared kmers with unused kmers from the model until all modifications were covered by unique kmers (see Code availability). For all kmers outside of modification branch points, we used the default RNA kmer distributions from ONT (r9.4_180 mv_70bps_5mer_RNA).

### Training configuration

SignalAlign v1.0.0 uses a variable-order hidden Markov model (HMM) which allows the number of paths through the HMM to be correctly constructed when ambiguous positions are defined (see Code availability) (**Rand et al., 2017**). Recent updates to signalAlign allow for relatively easy model definition and variant site selection which allows a user to define modifications, set prediction site locations and train a model. We defined all modified positions in the IVT sample as canonical and all positions in the wild type as modified. The locations of modified positions were determined by **Taoka et al., 2016** via mass spectrometry. For supervised training, all potential modified positions were defined as either canonical (IVT) or modified (wild type). Modifications within the first 15 nucleotides from the 5′ end of 18 S or 25 S cannot be called because the signal from that part of the sequence is not captured by standard nanopore sequencing. This limitation does not affect our experiments because there are no modifications in yeast which are within the first 15 nucleotides from the 5′ end of 18 S or 25 S.

We used 500 reads from both wild type and IVT reads and ran 30 rounds of training. For each round of training, we generated alignments between events and the reference sequence. Then, we

generated new event Gaussian distributions for all kmers covering modified positions. The mean of the gaussian distribution was defined as the median of the empirical kmer distribution and the standard deviation was defined as the median absolute deviation of the empirical kmer distribution. Similar to another study, we have seen that the median is less susceptible to being influenced by outliers (*Ding et al., 2020*). To train the model, we used `trainModels.py` from signalAlign (see Code availability).

## Inference and accuracy metrics

In order to test our results, we used 'runSignalAlign.py' and a trained model to predict modification status on all positions of 500 hold out IVT reads and 500 hold out wild-type reads (see Code availability). SignalAlign produced posterior probabilities of event alignments to both canonical kmers and modified kmers. We used `embed_main sa2bed` to decode the posterior probabilities from the signalAlign output into the probability of a position being modified (*Rand et al., 2017*). These probabilities are used for the receiver operating characteristic curve, precision-recall curve, calibration curve of *Figure 1—figure supplement 2*. A probability cutoff of 0.5 is used for the confusion matrix as well as the quantification of percent modified for any position.

We also compared accuracy on our test set to several snoRNA knockouts. Again, assuming snoRNA knockouts completely ablate target modifications and modifications are 100% present at all other positions, the average balanced accuracy over the snoRNA knockout positions is 85.1% and the expected balanced accuracy is 87.1% (*Supplementary file 1B,C*). Average balanced accuracy is calculated by getting the average of all balanced accuracies across all snoRNA knockout positions. Balanced accuracy for one position is calculated by adding the specificity to the sensitivity and dividing by two.

## Percent modification change

For every experiment and each modification position, we perform several chi-square two sample tests comparing the three wild type replicates modification frequencies to all experiment repeats' modification frequency (*Pearson, 1900*). We then select the highest p-value and correct for multiple tests using the Benjamani-Hochberg procedure (*Benjamini and Hochberg, 1995*). We also control for batch effects by filtering out positions which fall below the maximum change in modification frequency between the replicates of the wild type. All sample modification percentages, p-values from chi-square two sample tests compared to wild type replicates and IVT, and Benjamani-Hochberg corrected p-values can be found in (*Supplementary file 1B*).

## Hierarchical clustering analysis

### Dendrogram creation procedure

In order to determine any subclusters of reads based on a modification profile, we used hierarchical clustering on the modification profiles we generated from the inference step (*Pedregosa, 2011*; *Waskom, 2020*). We generated the dendrogram using Ward's method and euclidean distance as the distance metric (*Ward, 1963*). Before clustering analysis, we filter out reads which do not cover every modification site. UMAP dimension reduction was done using the umap python package and visualization using matplotlib (*Hunter, 2007*; *McInnes et al., 2018*).

### Cluster partitioning

To determine the number of reads in a set of N clusters we simply cut the dendrogram to create N subclusters and calculated the fraction of reads within each branch.

### Modification correlations

To calculate correlations between modified positions, we first filter out reads which did not cover all modifications and select the set of probabilities associated with each position. We then calculate the Spearman rank correlation between all pairwise combinations of modification positions on the same molecule. We use Spearman rank correlation to study the relationship between two probabilities because pairs of probabilities do not follow a bivariate normal distribution. p-Values were calculated using a two sided t-test and multiple tests corrected via the Benjamani-Hochberg procedure (*Benjamini and Hochberg, 1995*; *Student, 1908*).

To compare correlations between experiments, we used Fisher's z-transformation to convert correlations into z-scores and then performed a z-test to obtain p-values (*Fieller et al., 1957*; *Fieller and Pearson, 1961*; *Fisher, 1915*; *Zar, 2014*). We then correct for multiple tests using the Benjamani-Hochberg procedure (*Benjamini and Hochberg, 1995*). These p-values represent the confidence that, between two samples, there is a significant difference between the two correlations. All correlation plots have stars for positions which are both significantly different from a comparison experiment (wild type or IVT) and are significantly different from zero (p-value < 0.05). To account for variation in experimental repeats, we plot the minimum difference and highest corrected p-value for all pairwise comparisons between experimental repeats and wild-type repeats.

For higher order claims which require aggregating information from several hypothesis tests we use Empirical Brown's method (*Brown, 1975*; *Poole et al., 2016*). The Empirical Brown's method uses empirical data to calculate the covariance matrix which is used to extended Fisher's method to the dependent case by using a re-scaled $\chi^2$ distribution (see Code availability).

Spearman correlation values, original two sided t-test p-values, corrected two sided t-test p-values, Fisher z-transform test comparison p-values, and corrected Fisher z-transform tests p-values can be found in (*Supplementary file 1A*).

## Event visualization

Using a similar procedure outlined in a previous study (*Ding et al., 2020*), we gather the kmer to reference mapping generated from signalAlign and extract the most probable event to kmer alignment path using the maximum expected accuracy alignment (*Durbin et al., 1998*; *Rand et al., 2017*). For each read, we standardize the raw signal and calculate event means. Prior to clustering and visualization, we combine all reads together and standardize events by column. We generate the dendrogram using the same procedure as hierarchical clustering of modification profiles, Ward's method and euclidean distance (*Ward, 1963*).

For kmer distributions seen in *Figure 2—figure supplement 2*, we plot the kernel density estimate of all events aligning to the corresponding kmer with a probability greater than 0.5. We then simply plot the corresponding kmer distributions from the final trained kmer model.

## Sample compare site detection

### Tombo pipeline

Using Tombo version 1.5.1, initial embedding of fastq data into the raw fast5s was done with the `tombo preprocess annotate_raw_with_fastqs` and signal to reference alignment with `tombo resquiggle` (*Stoiber et al., 2016*). Finally, `tombo detect_modifications level_sample_compare` was used to generate windowed means of individual position Kolmogorov–Smirnov tests comparing the IVT sample position signal distributions to the wild-type sample (WT_YPD) position signal distributions (*Stoiber et al., 2016*). For a given position i, the windowed mean D-statistic is $w_i = \frac{\sum_{i-1}^{i+1} d_i}{3}$ where d is the D-statistic for a given position and w is the final reported statistic plotted in *Figure 1—figure supplement 1A, D*.

### Accuracy of modification site prediction

Using the D-statistic generated from Tombo, we calculated the per-position modification location detection AUROC (Area Under the Receiver Operating Characteristic) for yeast 18 S (0.924) and 25 S (0.934) rRNA (*Figure 1—figure supplement 1B, E*, see Supplementary Methods).

We also wanted to see if peaks corresponded with modifications, not just raw D-statistic values. So, we identified peaks and considered a peak within 2 nucleotides of a modification as a true positive. We then calculated the AUROC for this less stringent method on yeast 18 S (0.984) and 25 S (0.986) rRNA (*Figure 1—figure supplement 1C, F*, see Supplementary Methods).

### Modification labels and frequency

Underlying labels for modification and frequency for the *S. cerevisiae* 18 S and 25 S rRNA came from *Taoka et al., 2016*. Expected changes in modification frequency in the Dbp3 deletion experiment came from *Aquino et al., 2021*. SnoRNA modification sites on yeast rRNA come from the UMASS Amherst Yeast snoRNA database (*Piekna-Przybylska et al., 2007*).

## Data availability

Fastq files from all direct RNA sequencing runs and signalAlign modification calls are publicly available in NCBI's Gene Expression Omnibus (GEO) and are accessible through GEO Series accession number GSE186634 (https://www.ncbi.nlm.nih.gov/geo/query/acc.cgi?acc=GSE186634). Fast5 and fastq files for all direct RNA sequencing are available in the European Nucleotide Archive (ENA) at EMBL-EBI under accession number PRJEB48183 (https://www.ebi.ac.uk/ena/browser/view/PRJEB48183). A detailed description of the datasets used and sequenced in this work with their corresponding ENA or GEO IDs can be found in (*Supplementary file 1G*).

## Code availability

Documentation, install requirements, and analysis scripts for all work specific to this paper can be found at https://github.com/adbailey4/yeast_rrna_modification_detection (*Bailey, 2022a*). SignalAlign v1.0.0 can be found at https://github.com/UCSC-nanopore-cgl/signalAlign (*Bailey et al., 2022c*) and embed_fast5 1.0.0 can be found https://github.com/adbailey4/embed_fast5 (*Bailey, 2022b*).

## Supplementary methods

### Tombo pipeline

Using Tombo version 1.5.1, initial embedding of fastq data into the raw fast5s was done with the `tombo preprocess annotate_raw_with_fastqs` and signal to reference alignment with `tombo resquiggle` (*Stoiber et al., 2016*). Finally, `tombo detect_modifications level_sample_compare` was used to generate windowed means of individual position Kolmogorov–Smirnov tests comparing the IVT sample position signal distributions to the wild type sample (WT_YPD) position signal distributions (*Stoiber et al., 2016*). For a given position i, the windowed mean D-statistic is $w_i = \frac{i-1_i + 1}{d_i 3}$ where d is the D-statistic for a given position and w is the final reported statistic plotted in *Figure 1—figure supplement 1*.

### Accuracy of modification site prediction

In order to get a general view of how all of the modifications are affecting the current signal we analyzed the signal shift between in vitro transcribed (IVT) and one wild type sample (WT_YPD) using Tombo (*Stoiber et al., 2016*). The signal difference of 18 S and 25 S strands using Tombo is shown in Supplementary Fig. S1A and S1B, respectively. There is a clear correlation between annotated modified positions and signal deviation but in order to quantify the relative accuracy of both approaches, we naively labeled the per-position deviations with the corresponding windowed mean D-statistic. As shown in *Figure 1—figure supplement 1C, D*, the per-position modification calling detection AUROC (Area Under the Receiver Operating Characteristic) was 0.924 for 18 S and 0.934 for 25 S. However, if a canonical position is directly next to a modified position, it is very likely the underlying current is going to be shifted for that position. Also, the uncertainty of which specific nucleotide in the pore gives rise to the most significant signal shift makes site selection for kmer based sample compare frameworks very difficult (*Ding et al., 2020*; *Leger et al., 2019*; *Stoiber et al., 2016*). Therefore, instead of evaluating Tombo on the per-position modification calling accuracy, we used a less stringent metric of modification window calling accuracy.

We looked to see if a peak was within a window of a specific modification and disregarded large differences in signal in the neighboring 2 bases of a modification. Specifically, for each modification, we took the maximum corresponding statistic value of a window of 5 positions covering that modification. For example, if pos 20 was modified, the corresponding statistic for position 20 was the maximum value for positions 18, 19, 20, 21 and 22. Then, we removed the 2 upstream and downstream values from being classified. So, positions 18, 19, 21 and 22 will not be classified as true negatives or false positives. This approach allows for uncertainty of where the modification is within a small window of 5 positions and greatly reduces the false positive rate. As seen in *Figure 1—figure supplement 1C, D*, by decreasing the stringency of our accuracy metric we see a marked improvement of modification detection to an AUROC of 0.984 for 18 S and 0.986 for 25 S.

### Supplementary note

The most accurate prediction of partially modified, closely spaced clusters of nucleotides would employ models trained with all examples of such clusters, however such a collection of samples for

training does not exist and would be prohibitively expensive to create. Instead, we rely on prior information, experimental design expectations and signal comparisons to determine confidence in signalAlign predictions of modification clusters. Specifically, for high interest modification clusters, we validate modification profiles for clusters found using signalAlign by using nanopore signal patterns and clustering the underlying event means (see Materials and methods) (*Ding et al., 2020*). In *Figure 1—figure supplement 4*, we focus on interesting patterns of modifications positions located in the peptidyl transfer center (PTC) (Um2921, Gm2922, Ψ2923) and positions targeted by U24 (Cm1437, Am1449, Gm1450).

Prior to running our depletion experiments, we were uncertain if inhibiting box C/D snoRNP function would alter the modification status of Um2921 because both Um2921 and Gm2922 can be methylated with the non-snoRNP methyltransferase Sbp1 (*Lapeyre and Purushothaman, 2004*). However, we did expect the Cbf5 depletion would create a high proportion of reads with a modification pattern unseen by the model (only missing the Ψ2923). Thus, prior to analysis, we were uncertain on the number of high proportion modification patterns across these three positions. After analysis by signalAlign, we see similarly modified wild type and Nop58 depletion reads with a slight decrease in frequency of all three modifications in the Cbf5 depletion (*Figure 1—figure supplement 4A* and *Supplementary file 1B*). Thus, our initial hypotheses are that (1) Um2921 modification is not altered by inhibiting box C/D snoRNP function and (2) that the altered signal caused by missing Ψ2923 manifests as a slight ( < 5%) decrease in the predicted modification frequency of Um2921 and a larger ( < 10%) decrease of predicted modification frequency of Gm2922 and Ψ2923. To test this hypothesis, we used the underlying event means to identify the number of distinct signal means combinations through these PTC nucleotide positions. In the range of positions 2917–2922, we see two clear clusters; one cluster of IVT reads indicating three unmodified positions and one cluster with reads from the wild type and both depletion experiments (*Figure 1—figure supplement 4C*). Upon closer inspection of the most informative kmers (2921, 2922, 2923, and 2924), we see that the clustering of event means partitions Cbf5 depletion reads and 2'O-methyl depletion reads (*Figure 1—figure supplement 4E*). Given that we only see two main clusters different from the IVT cluster confirms that the 2'O-methyl depletion had little to no effect on modification status of Um2921 and Cbf5 depletion experiment most likely causes a slight decrease in modification at Ψ2923.

For the U24-dependent methylations near the PET, our model shows a high level of correlation between each position, and reads with unexpected missing 2'O-methyls at all three positions in the Cbf5 depletion, and reads with unexpected presence of 2'O-methyls at all three positions in the Nop58 depletion (*Figure 1—figure supplement 4A*). Given the isolation of Cm1437 from other nearby modified nucleotides, we are confident that the predicted states of Cm1437 modification in both depletion experiments are accurate. To test the model's predictions for the other two 2'O-methyls (Am1449 and Gm1450), we clustered the most informative kmers (1448, 1449, and 1450) (*Figure 1—figure supplement 4B*) and saw only two clusters of events, corresponding to unmodified IVT and fully modified wild type (*Figure 1—figure supplement 4D*). Given that we see no partitioning between the two depletion experiments confirms that there are only two primary modification patterns for Am1449 and Gm1450 in these data: either both are modified or both are unmodified. This analysis indicates that (1) 2'O methylation at these three positions is highly concerted, (2) depletion of the pseudouridylase Cbf5 leads to a decrease in U24 2'O methylation efficiency and (3) these three positions together can remain modified in the Nop58 depletion, or a required for stability of rRNA, or both.

## Acknowledgements

We thank our colleagues for graciously sharing strains and plasmids: Skip Fournier and Wayne Decatur (snR80, snR83, and snR87 deletions), David Tollervey (*GAL-CBF5* and *GAL-NOP58*), Raymond O'Keefe (snR4 and snR45 deletions), Jon Staley (*prp43-Q423N*), Brian Rymond (*spp382-1, pxr1Δ*), Hiten Madhani (*prp16-302*), Jelena Micic and John Woolford (rDNA plasmid originally provided by Skip Fournier). We also thank Nicholas Forino and the Michael Stone lab for initial use of the flongle adaptor. We also thank Jordan Eizenga, John Paul Donohue, Miten Jain, and Logan Mulroney for technical assistance and advice, and to colleagues who provided scientific discussion and feedback on earlier versions of this manuscript: Gina Mawla, Guillaume Chanfreau, Jelena Micic, Idil Ulengin-Talkish, Jen Quick-Cleveland and John Woolford Jr.

This research was supported by NIH grants R01 HG010053 (to M Akeson, Ares and Paten Co-PIs) and R01 GM040478 (M Ares), as well as U41HG010972, R01HG010485, U01HG010961, OT3HL142481, OT2OD026682, U01HL137183, and 2U41HG007234 to B Paten.

## Additional information

### Funding

| Funder | Grant reference number | Author |
|---|---|---|
| National Institute of General Medical Sciences | R01 GM040478 | Manuel Ares |
| National Human Genome Research Institute | R01 HG010053 | Manuel Ares |
| National Human Genome Research Institute | U41HG010972 | Benedict Paten |
| National Human Genome Research Institute | R01HG010485 | Benedict Paten |
| National Human Genome Research Institute | U01HG010961 | Benedict Paten |
| NIH Office of the Director | OT2OD026682 | Benedict Paten |
| National Heart, Lung, and Blood Institute | U01HL137183 | Benedict Paten |
| National Human Genome Research Institute | 2U41HG007234 | Benedict Paten |

The funders had no role in study design, data collection and interpretation, or the decision to submit the work for publication.

### Author contributions

Andrew D Bailey, Conceptualization, Data curation, Formal analysis, Investigation, Methodology, Software, Validation, Visualization, Writing – original draft, Writing – review and editing; Jason Talkish, Conceptualization, Data curation, Formal analysis, Investigation, Methodology, Resources, Validation, Visualization, Writing – original draft, Writing – review and editing; Hongxu Ding, Methodology, Software, Validation; Haller Igel, Investigation; Alejandra Duran, Shreya Mantripragada, Software; Benedict Paten, Conceptualization, Data curation, Formal analysis, Funding acquisition, Investigation, Methodology, Project administration, Software, Supervision, Validation, Visualization, Writing – original draft, Writing – review and editing; Manuel Ares, Conceptualization, Formal analysis, Funding acquisition, Investigation, Methodology, Project administration, Resources, Supervision, Validation, Visualization, Writing – original draft, Writing – review and editing

### Author ORCIDs

Andrew D Bailey ⓘ http://orcid.org/0000-0001-8304-7565
Jason Talkish ⓘ http://orcid.org/0000-0002-0260-3345
Shreya Mantripragada ⓘ http://orcid.org/0000-0002-6234-6176
Manuel Ares ⓘ http://orcid.org/0000-0002-2552-9168

### Decision letter and Author response

Decision letter https://doi.org/10.7554/eLife.76562.sa1
Author response https://doi.org/10.7554/eLife.76562.sa2

## Additional files

### Supplementary files

• Supplementary file 1. Supplementary tables. (A) Spearman correlations between reference positions along with corresponding p-value, Benjamini-Hochberg corrected p-values, correlation

comparison Fisher z-transform test, Benjamini-Hochberg corrected correlation comparison Fisher z-transform test p-values with wild type repeats and IVT. (B) Fraction modified by position for every experiment with associated wild type- to-experiment two sample chi squared test p-values and Benjamini-Hochberg corrected p-values. (C) Fraction modified by position, average fraction modified and standard deviation of the three replicates for wild type yeast rRNA. (D) Yeast strains used in this study. (E) Oligonucleotides used in this study. (F) Sequencing metrics for *S. cerevisiae* rRNA using direct RNA nanopore sequencing. (G) Experiment mapping between ENA, SRA, GEO IDs.

- Transparent reporting form

### Data availability

Fastq files from all direct RNA sequencing runs and signalAlign modification calls are publicly available in NCBI's Gene Expression Omnibus (GEO) and are accessible through GEO Series accession number GSE186634 (https://www.ncbi.nlm.nih.gov/geo/query/acc.cgi?acc=GSE186634). Fast5 and fastq files for all direct RNA sequencing are available in the European Nucleotide Archive (ENA) at EMBL-EBI under accession number PRJEB48183 (https://www.ebi.ac.uk/ena/browser/view/PRJEB48183). A detailed description of the datasets used and sequenced in this work with their corresponding ENA or GEO IDs can be found in (Supplementary file 1G). Code availability Documentation, install requirements, and analysis scripts for all work specific to this paper can be found at https://github.com/adbailey4/yeast_rrna_modification_detection. SignalAlign v1.0.0 can be found at https://github.com/UCSC-nanopore-cgl/signalAlign and embed_fast5 1.0.0 can be found https://github.com/adbailey4/embed_fast5.

The following datasets were generated:

| Author(s) | Year | Dataset title | Dataset URL | Database and Identifier |
|---|---|---|---|---|
| Bailey AD, Talkish J, Ding H, Igel H, Duran A, Mantripragada S, Paten B | 2021 | Concerted modification of nucleotides at functional centers of the ribosome revealed by single-molecule RNA modification profiling | https://www.ncbi.nlm.nih.gov/geo/query/acc.cgi?acc=GSE186634 | NCBI Gene Expression Omnibus, GSE186634 |
| Bailey AD, Talkish J, Ding H, Igel H, Duran A, Mantripragada S, Paten B | 2021 | Concerted modification of nucleotides at functional centers of the ribosome revealed by single-molecule RNA modification profiling | https://www.ebi.ac.uk/ena/browser/view/PRJEB48183 | European Nucleotide Archive, PRJEB48183 |

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
