## [Editor Report]

The paper describes a method for single-molecule profiling of RNA modifications. The results not only solve many urgent questions in understanding rRNA modification, ribosome heterogeneity and ribosome biogenesis, they also provide a major step in developing technologies to probe the RNA epitranscriptome. The results are expected to be of broad interest for specialists in the RNA field.

---

## [Decision Letter]

**Decision letter after peer review:**

Thank you for submitting your article "Concerted modification of nucleotides at functional centers of the ribosome revealed by single-molecule RNA modification profiling" for consideration by *eLife*. Your article has been reviewed by 3 peer reviewers, including Marina V Rodnina as the Reviewing Editor and Reviewer #1, and the evaluation has been overseen by James Manley as the Senior Editor. The following individuals involved in review of your submission have agreed to reveal their identity: Katrin Karbstein (Reviewer #2); Guillaume F Chanfreau (Reviewer #3).

Essential revisions:

1. The experiments with the helicase mutants may need additional controls. The modification pattern obtained with the cold-sensitive prp44 should be compared to that with the wt after a shift to the same non-permissive temperature. It would be necessary to perform a control experiment with a wild-type strain shifted to a similar temperature. Also, the dbp knockout analysis is performed at steady state while prp43-cs is a cold shift so it is quite difficult to compared the result directly.

2. The conclusion concerning the resilience of modification to stress should be supported by some considerations concerning the potential time frame of stress response vs. the time of synthesis of new ribosomes that could potentially have a different modification level. Regarding splicing perturbations, and with the exception of the dbr1 knockout, the mutants used in the study do not result in a major depletion of intron encoded snoRNAs so it is quite expected that there is no loss of modification at these positions. Similarly, the environmental stresses used are short, and are not expected to affect modification patterns in a major way considering the stability of ribosomes. Unless the authors perform sequencing on rRNAs synthesized after a shift into stress conditions, it is misleading to state that rRNA modification profiles are unaffected by environmental treatments.

3. Another issue that may need to be considered is the level of depletion of individual snoRNAs after depletion of the snoRNP proteins. It is possible that some snoRNAs are depleted more rapidly than others, and that this may affect the modification patterns. The authors should perform RNA sequencing of RNA samples used after depletion of Cbf5 or Nop58 such that they can directly correlate snoRNA levels to modification levels. Unless the authors provide these data, it is difficult to conclude whether specific sites are more or less resilient to genetic depletion of snoRNP proteins.

4. On p.10, the authors benchmark their method in challenging, hypermodified regions. More generally, modifications close to each other should be contained in some conventional datasets. As these are mostly correlated (or anticorrelated) some attempts to identify that in the existing data should be made.

5. The authors should analyze the correlated and anticorrelated sites in more depth and in particular consider the RNA secondary structure, as the base pairing to snoRNAs disrupts that (as well as RP binding). In fact, in a first look, it seems to me that the anticorrelated changes could be explained by competing binding of two distinct snoRNAs.

6. Why there are so few ribosomes reads in each sample. A typical Oxford Nanopore flow cell will typically yield ~1 million reads and yet it seems like only a few hundred make it into the figure.

7. There are definitely some hot spots for miscalled modifications in the IVT samples. Would be interesting to hear the author's thoughts on why these sights are miscalled. They mention a few examples later in the text, but a more thorough discussion is needed.

8. Non-full-length ribosomes are discarded. It's possible that 2'O-met may increase the stability of ribosomes by preventing inline hydrolytic cleavage. This may bias the presented data to 2'O-met modifications. This could be discussed.

9. The heatmaps are somewhat difficult to interpret. In all figures, the heatmaps could be enlarged and the cluster matrix and UMAP projections could be deleted or made smaller.

10. Figure 2D snR80 target should be 25S – Ψ766.

11. P5 – Line 112 "which either" needs rewording.

12. – Resistant of Um2921 of GM2922 to depletion of Nop58. The authors conclude this paragraph by saying that the "low number of ribosomes unmodified at these positions suggest that their modification may be essential for rRNA stability". This conclusion is incorrect, considering that Spb1 can methylate both Gm2922 and Um2921 when snR52 is depleted. So, the absence of unmodified rRNA upon depletion of Nop58 (or Cbf5) can be fully explained by the functional redundancy with Spb1.

13. "to a large extent, 2´O-methylation is independent of pseudouridylation in yeast rRNA (Figure 1D and 1E).": I would suggest changing the wording to " 2´O-methylation and pseudouridylation in yeast rRNA are independent of each other".

14. Figure 4 is incredibly hard to understand. I just could not grasp the information presented. I suggest that the authors find an alternative way to present the information.

15. P20 line 368. I do not think that the reference Piekna-Przybylska 2007 is correctly cited here. This paper is about "new bioinformatics package for research on rRNA nucleotide modifications in the ribosome" and it does not provide information related to the fact that synthesis of intron encoded snoRNAs is compromise d by mutations that affect splicing.

16. P21 – line 390. Alternative snoRNA maturation pathways independent of splicing. The reference Grzechnick et al. 2018 is not correct here, as this paper deals with independently transcribed snoRNAs that are not encoded within introns. Correct reference should be Ooi et al. 1998 or papers from the Bozzoni group that showed splicing independent processing of intron encoded snoRNAs in yeast.

*Reviewer #1 (Recommendations for the authors):*

1. The experiments with the helicase mutants may need additional controls. For example, the modification pattern obtained with the cold-sensitive prp44 should be compared to that with the wt after a shift to the same non-permissive temperature.

2. The conclusion concerning the resilience of modification to stress should be supported by some considerations concerning the potential time frame of stress response vs. the time of synthesis of new ribosomes that could potentially have a different modification level.

*Reviewer #2 (Recommendations for the authors):*

There are some ways the manuscript could be improved:

1. On p.10, the authors benchmark their method in challenging, hypermodified regions. More generally, modifications close to each other should be contained in some conventional datasets. As these are mostly correlated (or anticorrelated) some attempts to identify that in the existing data should be made.

2. I would suggest that the authors analyze the correlated and anticorrelated sites in more depth and in particular consider the RNA secondary structure, as the base pairing to snoRNAs disrupts that (as well as RP binding). In fact, in a first look, it seems to me that the anticorrelated changes could be explained by competing binding of two distinct snoRNAs…

*Reviewer #3 (Recommendations for the authors):*

I'm confused about why there are so few ribosomes reads in each sample. A typical Oxford Nanopore flow cell will typically yield ~1 million reads and yet it seems like only a few hundred make it into the figure.

There are definitely some hot spots for miscalled modifications in the IVT samples. Would be interesting to hear the author's thoughts on why these sights are miscalled. They mention a few examples later in the text, but a more thorough discussion is needed.

Non-full-length ribosomes are discarded. It's possible that 2'O-met may increase the stability of ribosomes by preventing inline hydrolytic cleavage. This may bias the presented data to 2'O-met modifications. This could be discussed.

The heatmaps are somewhat difficult to interpret. In all figures, the heatmaps could be enlarged and the cluster matrix and UMAP projections could be deleted or made smaller.

Figure 2D snR80 target should be 25S – Ψ766.

P5 – Line 112 "which either" needs rewording.

– Resistant of Um2921 of GM2922 to depletion of Nop58. The authors conclude this paragraph by saying that the "low number of ribosomes unmodified at these positions suggest that their modification may be essential for rRNA stability". This conclusion is incorrect, considering that Spb1 can methylate both Gm2922 and Um2921 when snR52 is depleted. So the absence of unmodified rRNA upon depletion of Nop58 (or Cbf5) can be fully explained by the functional redundancy with Spb1.

"…to a large extent, 2´O-methylation is independent of pseudouridylation in yeast rRNA (Figure 1D and 1E).": I would suggest changing the wording to " 2´O-methylation and pseudouridylation in yeast rRNA are independent of each other".

Figure 4 is incredibly hard to understand. I just could not grasp the information presented. I suggest that the authors find an alternative way to present the information.

P20 line 368. I do not think that the reference Piekna-Przybylska 2007 is correctly cited here. This paper is about "new bioinformatics package for research on rRNA nucleotide modifications in the ribosome" and it does not provide information related to the fact that synthesis of intron encoded snoRNAs is compromise d by mutations that affect splicing.

P21 – line 390. Alternative snoRNA maturation pathways independent of splicing. The reference Grzechnick et al. 2018 is not correct here, as this paper deals with independently transcribed snoRNAs that are not encoded within introns. Correct reference should be Ooi et al. 1998 or papers from the Bozzoni group that showed splicing independent processing of intron encoded snoRNAs in yeast.

---

## [Author Response]

Essential revisions:1. The experiments with the helicase mutants may need additional controls. The modification pattern obtained with the cold-sensitive prp44 should be compared to that with the wt after a shift to the same non-permissive temperature. It would be necessary to perform a control experiment with a wild-type strain shifted to a similar temperature. Also, the dbp knockout analysis is performed at steady state while prp43-cs is a cold shift so it is quite difficult to compared the result directly.

We did perform this control in the original manuscript (it is critical for the analysis!) but failed to contrast it to the experimental sample in our presentation. We have now remedied that oversight. The reviewer’s first concern is for a cold shifted wt sample to control for cold shift effects not related to *prp43*’s phenotype, and the second is how one compares a steady state isothermal phenotype of one mutant to a cold shifted phenotype of a second mutant. Regarding the first point, we have now made new supplemental figures (Figure 3 —figure supplements 2 and 3) in which wt with and without cold shift are compared directly with cold shifted *prp43-cs*, and a control cs splicing mutant *prp16-cs*. We had previously documented the lack of change in modification in wt upon cold shift in Figure 5, but the new figure shows this directly alongside the two cs helicase mutations used in this study, reinforcing our original conclusion that *prp43-cs* has a specific defect not generated by either cold shift (WT 18 degrees) or splicing inhibition (*prp16-cs*).

Regarding the second point, we wanted to be careful to compare each mutant with wild type under the same conditions, identify the defects of each relative to wild type, and then compare those. It is a general problem in genetics that conditional phenotypes maybe not be compared directly to each other if the condition that generates the phenotype is different, and the best solution is to control for the condition. It is also true that a steady state phenotype may be different than a shift-induced phenotype due to ongoing ancillary changes in gene expression during the shift. Here we’re not in an optimal situation, as a *prp43* null is inviable and cannot be compared to a dbp3 null, and we do not have a cold sensitive *dbp3* mutant. Happily, the phenotypes are qualitatively different, and we already know the two proteins do not carry out the same exact function because they don’t complement each other. A deeper analysis of any overlap between the functions of Prp43 and Dbp3 would require additional experiments, for example an epistasis test, however those experiments would be best done as part of a follow up study. The text changes that describe the new figure with the controls directly next to each mutant are on page 15, line 361.

2. The conclusion concerning the resilience of modification to stress should be supported by some considerations concerning the potential time frame of stress response vs. the time of synthesis of new ribosomes that could potentially have a different modification level.

Here we could have been clearer about the motivation for our experiments. Throughout the study, we used 1 hour as a standard treatment time to represent an acute change in environmental conditions, both for consistency and because many of the treatments are not particularly growth inhibitory. At 30°C for example, cell numbers increase about 1.6 fold per hour (doubling time is ~100 min). We expected ribosomes to accumulate along a similar path, meaning that after 1 hr, as much as 38% or so of the existing ribosomes (to a first approximation) might have been newly synthesized and modified under the new conditions, and thus detectable if the treatment only affected new ribosomes. In that case the loss or gain of a modification as an immediate response to treatment would have appeared in a substantial minority of the ribosomes after 1 hour if such modifications existed. Our inability to detect such modifications does not mean there is no effect of stress on modification over longer time scales, and we have addressed this by clarifying the presentation of the experiment on pages 19-20, lines 465-492.

Regarding splicing perturbations, and with the exception of the dbr1 knockout, the mutants used in the study do not result in a major depletion of intron encoded snoRNAs so it is quite expected that there is no loss of modification at these positions. Similarly, the environmental stresses used are short, and are not expected to affect modification patterns in a major way considering the stability of ribosomes. Unless the authors perform sequencing on rRNAs synthesized after a shift into stress conditions, it is misleading to state that rRNA modification profiles are unaffected by environmental treatments.

As described above, we do not make any claims about the long-term effect of our 1-hour treatments, as we have not measured modification over the longer term under those conditions. We looked for effects of splicing-related mutations because of known connections between splicing and ribosome biogenesis: (1) 90% of the splicing done in vegetatively growing yeast is devoted to the translation apparatus, (2) some snoRNAs are intron-encoded, and (3) Prp43 has roles in both ribosome biogenesis and splicing. Reduction of snoRNA levels might or might not be expected, however this is not the only possible mechanism for loss of modification. We have clarified this by adding: “Loss of rRNA modifications in response to splicing, environment or stress conditions could occur through at least two mechanisms: 1) enzymatic removal of pre-existing modifications or 2) synthesis of nascent rRNAs that lack snoRNA-guided rRNA modifications.” on page 20, lines 480-492 of the revised manuscript. Furthermore, we have clarified throughout this manuscript that these are “acute” stressors.

Ultimately the splicing mutants represent important controls for Prp43 activity in ribosome biogenesis. We used 4 splicing-related mutants, *dbr1* deletion (causes lariat accumulation with subtle effects on levels of intronic snoRNAs but no major inhibition of splicing), *spp382-1* (G patch protein for Prp43 function in splicing, inhibits splicing termination), *prp16-302* (cold sensitive second step factor that accumulates lariat intermediates) and *prp43-Q423N* (cold sensitive for splicing termination and ribosome biogenesis). The first three mutants might have caused a loss of modification at sites guided by intronic snoRNAs, and were used to exclude splicing defects of *prp43-Q423N* as an explanation for *prp43*-dependent loss of rRNA modification.

3. Another issue that may need to be considered is the level of depletion of individual snoRNAs after depletion of the snoRNP proteins. It is possible that some snoRNAs are depleted more rapidly than others, and that this may affect the modification patterns. The authors should perform RNA sequencing of RNA samples used after depletion of Cbf5 or Nop58 such that they can directly correlate snoRNA levels to modification levels. Unless the authors provide these data, it is difficult to conclude whether specific sites are more or less resilient to genetic depletion of snoRNP proteins.

This is a very interesting experiment that we did not consider, but we are concerned that the wholesale depletion of the Nop58 or Cbf5 proteins of snoRNPs may be too complex to obtain reliable information about the complex relationships between snoRNA levels and modification efficiency across hundreds of modified sites. Steady state levels of snoRNAs appear to vary by more than 10-fold in wt cells despite resulting in equivalent levels of modification, suggesting that snoRNA level per se may not be strictly coupled to modification efficiency. In the two decades old Nop58 and Cbf5 depletion experiments we reproduced from Lafontaine, Tollervey and colleagues, snoRNAs are also likely competing for increasingly smaller amounts of protein, and relative amounts of residual snoRNA may not be assembled, obscuring the connection between snoRNA level and functional snoRNP level. We do not believe that the endpoint, catastrophic depletion experiment can provide much quantitative information about the relative activities of snoRNPs within a class but remain confident in the qualitative differences observed for the two distinct snoRNA classes.

4. On p.10, the authors benchmark their method in challenging, hypermodified regions. More generally, modifications close to each other should be contained in some conventional datasets. As these are mostly correlated (or anticorrelated) some attempts to identify that in the existing data should be made.

We thank the reviewer for their comment. Unfortunately, to our knowledge, no published data exist in which subclasses of partly modified fragments containing closely spaced modifications have been described. The mass spectrometry study from Taoka et al. 2016 identified and quantified several fragments with multiple modifications but only reported individual modification frequencies, usually >95%, suggesting that residual, partly modified fragments were not detected or could not be analyzed for modification status at the nearby site. Modification correlation analysis was also not done in HPLC modification papers (Yang et al. 2016), RiboMeth-Seq papers (Birkedal et al. 2014; Marchand et al. 2016), or primer extension-based papers. We did confirm the concerted loss of modification pattern by comparing the nanopore signal means directly.

5. The authors should analyze the correlated and anticorrelated sites in more depth and in particular consider the RNA secondary structure, as the base pairing to snoRNAs disrupts that (as well as RP binding). In fact, in a first look, it seems to me that the anticorrelated changes could be explained by competing binding of two distinct snoRNAs.

We are excited that the reviewer believes as we do that there is much in the data that has not fully been explored! The reviewer is imagining scenarios in which binding of a snoRNA has structural consequences on partly assembled ribosomes that influence rRNA folding or ribosomal protein binding at distant locations, that then affects the access of a second snoRNP to its substrate. There are numerous situations during ribosome biogenesis where this sort of dependency or collision could occur, including the example we detailed concerning *dbp3* and *prp43* in Figure 6. To pursue more thoroughly the specific relationships suggested by the reviewer between distinct snoRNAs that may bind in a mutually exclusive way to the pre-ribosome we would prefer to test a comprehensive set of single snoRNA knockout strains. However, in response to the reviewer’s question, we searched for correlation changes between snoRNA knockouts we have already performed and discovered a relationship between snR83 and snR4 and their targets. This is now described in a new figure (Figure 2—figure supplement 3) and with a short additional section (pages 12-13, lines 297-316). We thank the reviewer for their insight and enthusiasm and hope to address this question in more detail with additional experiments in the future.

6. Why there are so few ribosomes reads in each sample. A typical Oxford Nanopore flow cell will typically yield ~1 million reads and yet it seems like only a few hundred make it into the figure.

The reviewer is correct that a typical Oxford Nanopore flow cell can yield ~1 million reads. Our initial experiments with IVT rRNA and the Nop58 and Cbf5 depletion strains were run on MinION flow cells and yielded ~171k-898k reads, ~71k-264k of which were full-length reads after filtering. To increase the throughput and lower the cost of these experiments, we developed a protocol to directly sequence rRNAs on Flongle flow cells that produced an average of ~5.9k full-length 18S and 25S rRNA reads per experiment. While full analysis was done on all full-length reads obtained from each experiment, the hierarchical clustering heatmaps throughout the paper display 500-1000 randomly selected reads to keep the figures from growing even larger and reducing their clarity. Total aligned reads and full-length read numbers for each experiment are reported in table supplement S6. We have also indicated within the figure legends the number of reads used to generate the heatmaps in each figure. Furthermore, we have added text to the manuscript on page 5, line 132 and page 10, lines 259-261 to indicate where we used MinION flow cells and Flongle flow cells.

7. There are definitely some hot spots for miscalled modifications in the IVT samples. Would be interesting to hear the author's thoughts on why these sights are miscalled. They mention a few examples later in the text, but a more thorough discussion is needed.

We thank the reviewer for raising this issue. Modifications with a small signal to noise ratio are less accurate when compared to other modifications. One measure of the signal to noise is the distance between the canonical and modified kmer distributions. The plot in Author response image 1 shows each modification’s specificity (IVT accuracy) compared to the maximum distance between kmer distributions.

**Author response image 1. sa2fig1:** 

The very minor positive correlation implies that the maximum distance between kmer distributions plays a small role in the variation of accuracy but there is probably variation not captured by the kmer distributions. Unfortunately, it is not obvious why some positions have a lower accuracy vs others without a thorough exploration of the differences between signals of specific modifications in specific contexts. One possibility is that there are other signal statistics, such as event duration, which are more important for lower accuracy modifications. Some of these ideas have been added to the manuscript on page 28, lines 670-676. However, the data exploration required to resolve this question would require additional technical development that likely would not affect the biological conclusions.

8. Non-full-length ribosomes are discarded. It's possible that 2'O-met may increase the stability of ribosomes by preventing inline hydrolytic cleavage. This may bias the presented data to 2'O-met modifications. This could be discussed.

The reviewer raises a point that we did not consider, because we were focused on full length single molecules. We acknowledged that we may be missing turnover products that may arise in the absence of modifications in the Results section on pages 8-9 lines 212-219. We also acknowledge in the Discussion on page 29, lines 687-691 that capturing and characterizing the modification status of potential turnover products will be necessary for understanding the full impact of modifications on ribosome biogenesis. It will be interesting to determine the full effect of modification loss on rRNA stability, but this might require a method to capture cleaved ends specifically.

9. The heatmaps are somewhat difficult to interpret. In all figures, the heatmaps could be enlarged and the cluster matrix and UMAP projections could be deleted or made smaller.

We appreciate the difficulty in presenting such dense and complex data in a simple visual format and thank the reviewer for this suggestion. As it is, images of one or the other molecule (18 or 25S) has been relegated to the supplement due to the presentational challenges. We could have split the molecules into shorter windows, but we believe this reduces the impact of the results. We have decided to include both the heatmaps and UMAP projections. In all the figures we present, we have tried to maintain sufficient resolution to allow enlargement on a screen for inspection of fine detail.

10. Figure 2D snR80 target should be 25S – Ψ766.

After carefully checking the current literature, we believe that 25S Ψ776 is the correctly annotated target of snR80 and it is labeled as such in Figure 2D. Although it was previously annotated as Ψ775, multiple current annotations reference it as Ψ776, including the UMass Amherst Yeast snoRNA Database (https://people.biochem.umass.edu/sfournier/fournierlab/snornadb/mastertable.php).

11. P5 – Line 112 "which either" needs rewording.

Thank you to the reviewer for helping us clarify this important point. We have changed the text on page 5, lines 144-147 to “To test the ability of the trained model to capture single molecule modification profiles, we examined yeast suffering catastrophic loss of snoRNA-guided rRNA modifications, by depleting either the C/D box (2´O-methylation) or H/ACA box (pseudouridylation) class of snoRNPs.”

12. – Resistant of Um2921 of GM2922 to depletion of Nop58. The authors conclude this paragraph by saying that the "low number of ribosomes unmodified at these positions suggest that their modification may be essential for rRNA stability". This conclusion is incorrect, considering that Spb1 can methylate both Gm2922 and Um2921 when snR52 is depleted. So, the absence of unmodified rRNA upon depletion of Nop58 (or Cbf5) can be fully explained by the functional redundancy with Spb1.

Thank you to the reviewer for helping us clarify our conclusion – we did mean the modification here not necessarily just snR52. It is true that Spb1 provides a redundant mechanism to methylate Um2921 and Gm2922, suggesting these modifications are critical for ribosome function. We have added a citation for a recently posted bioRxiv preprint from Arlen Jonson’s lab that shows modification of Gm2922 is required for efficient nuclear export and maturation of the pre-60S subunit (Yelland et al. 2022). We also did not previously make clear that there is no known redundant snoRNP or enzymatic mechanism for pseudouridylation of U2923. We have added text to this point. However, we still think it reasonable to conclude that the loss of one or more of this triad of modifications, including any molecules that may escape a redundant mechanism, are unlikely to be exported to the cytoplasm, or otherwise become unstable, explaining their absence in the data. All these changes can be found on pages 8-9, lines 212-219.

13. "to a large extent, 2´O-methylation is independent of pseudouridylation in yeast rRNA (Figure 1D and 1E).": I would suggest changing the wording to " 2´O-methylation and pseudouridylation in yeast rRNA are independent of each other".

We thank the reviewer for their suggestion to clarify this important observation and have edited the manuscript accordingly.

14. Figure 4 is incredibly hard to understand. I just could not grasp the information presented. I suggest that the authors find an alternative way to present the information.

We thank the reviewer for their feedback regarding Figure 4. However, after carefully considering how to present this complex and novel data set, we have decided to keep it as it is. Throughout this project we have considered and tried various approaches to presenting this data in two- and three-dimensional space. While we agree that it is a complex figure that requires a bit of time to properly digest, we think that a graphical representation of modifications where edges represent a change in correlation between two positions is a good way to visualize the groups of modifications that share information. Furthermore, it allows us to visualize how positions within functional centers of the ribosome change together or independently. We think this is a good way to keep track of these relationships, however we would caution that they may not be immediately interpretable in terms of molecular processes that we think might take place during ribosome biogenesis. We have added some information to the figure legend to try to clearly state what we hope the reader will focus on when viewing this figure.

15. P20 line 368. I do not think that the reference Piekna-Przybylska 2007 is correctly cited here. This paper is about "new bioinformatics package for research on rRNA nucleotide modifications in the ribosome" and it does not provide information related to the fact that synthesis of intron encoded snoRNAs is compromise d by mutations that affect splicing.

We thank the reviewer for pointing this out and have fixed the text.

16. P21 – line 390. Alternative snoRNA maturation pathways independent of splicing. The reference Grzechnick et al. 2018 is not correct here, as this paper deals with independently transcribed snoRNAs that are not encoded within introns. Correct reference should be Ooi et al. 1998 or papers from the Bozzoni group that showed splicing independent processing of intron encoded snoRNAs in yeast.

We thank the reviewer for pointing out this mistake and we changed the reference to Processing of the intron-encoded U18 small nucleolar RNA in the yeast *Saccharomyces cerevisiae* relies on both exo- and endonucleolytic activities. Villa T, Ceradini F, Presutti C, Bozzoni I. (Villa 1998).